# Predicting IVF outcomes using a logistic regression–ABC hybrid model: A proof-of-concept study on supplement associations

Uğur Ejder[1]*, Pınar Uskaner Hepsağ[2]

1 Department of Information Technology, Adana Alparslan Türkeş Science and Technology University, Adana, Türkiye, 2 Faculty of Computer and Informatics, Adana Alparslan Türkeş Science and Technology University, Adana, Türkiye

* uejder@atu.edu.tr

## Abstract

Machine learning models are increasingly applied to assisted reproductive technologies (ART), yet most studies rely on conventional algorithms with limited optimization. This proof-of-concept study investigates whether a hybrid Logistic Regression–Artificial Bee Colony (LR–ABC) framework can enhance predictive performance in in vitro fertilization (IVF) outcomes while producing interpretable, hypothesis-driven associations with nutritional and pharmaceutical supplement use. A retrospective dataset of 162 women undergoing IVF was analyzed. Clinical, demographic, and supplement variables were preprocessed into 21 predictors. Four algorithms (K-Nearest Neighbors, Classification and Regression Tree, Support Vector Machine, and Random Forest) were implemented alongside their LR–ABC hybrid counterparts. Model performance was evaluated using 5-fold cross-validation with Synthetic Minority Over-sampling Technique (SMOTE) to address class imbalance. Local Interpretable Model-agnostic Explanations (LIME) were applied to improve interpretability. Across all algorithm models, LR–ABC hybrids outperformed their baseline models (e.g., Random Forest: 85.2% → 91.36% accuracy). LIME explanations identified omega-3, folic acid, and dietician support as influential features in individual predictions. However, given the small sample size, binary representation of supplements, and absence of external validation, the observed improvements and associations should be regarded as exploratory rather than definitive. The LR–ABC hybrid model demonstrates methodological potential for improving prediction and interpretability in IVF research. Findings regarding supplement associations are hypothesis-generating, not clinically directive. Future studies with larger, multi-center datasets including detailed dosage and dietary data are needed to validate and extend this framework.

**Data availability statement:** https://github. com/ugurejder/ABC_IVF/blob/main/IVF_ english_dataset.xlsx.

**Funding:** This study was supported by the Scientific and Technological Research Council of Turkey (TÜBİTAK) through a publication incentive program. TÜBİTAK had no role in the study design, data collection and analysis, decision to publish, or preparation of the manuscript.

**Competing interests:** The authors have declared that no competing interests exist.

**Abbreviations:** ABC, Artificial bee colony; ART, Aided reproductive technologies; CART, Classification And Regression Tree; DHA, Docosahexaenoic Acid; IVF, In vitro fertilization; KNN, K-Nearest Neighbors; ML, Machine Learning; SVM, Support Vector Machines; WHO, World Health Organization; Acc, Accuracy.

## 1. Introduction

Subfertility is defined as the inability to become pregnant after one year of regular, unprotected sexual intercourse. According to the World Health Organization (WHO), about 10–15% of couples worldwide are affected by subfertility (World Health Organization, 2020). As the world's population ages, subfertility is becoming an increasingly common problem and one of the biggest public health challenges of the 21st century [1]. The trend towards postponing childbearing, particularly in high-income countries, has contributed to the rise in subfertility cases and further increased the demand for effective treatment solutions [2]. undertook a review of future directions for subfertility research and intervention. His study focused on three key areas: helping and offering alternatives to infertile people, targeting avoidable causes of subfertility, and boosting new LCIVF initiatives to improve the availability and acceptability of global aided reproductive technologies (ART).

There are various approaches to treating subfertility, including pharmacological therapies, surgical interventions, and ART such as in vitro fertilization (IVF) and intrauterine insemination (IUI) [3]. However, these treatment methods are often costly, emotionally draining, and physically demanding, making the subfertility treatment process both stressful and lengthy [3]. [4] provide a comprehensive overview of subfertility, how it's defined, and how it affects the world.

The aim of Aoun's review was to synthesize recent findings on the influence of nutritional factors, including specific food groups, nutrients, and dietary supplements, on sexual and reproductive function in both men and women. This effort allowed for the discovery of relevant studies that provide insight into the potential role of nutrition in reproductive health [5]. Dietary education for infertile women is an important way to improve their awareness and treatment outcomes. With the growing use of smartphones, the design of a mobile-based nutrition education application for women with subfertility problems can be of great benefit according to the cultural conditions [6]. [7] present a machine learning model to detect the most relevant bioactive molecules and clinical drugs associated with genes underlying pathways known to be significant in predisposition to polycystic ovary syndrome [7,8]. [9] are reviewing existing data on how supplements, diet, and lifestyle changes affect weight and how they affect fertility in both men and women [9].

One promising solution is the incorporation of computer-aided diagnostic models into clinical practice. These models can significantly improve the accuracy of predicting subfertility treatment outcomes, which can lead to higher success rates [10,11]. By helping clinicians make more informed decisions, these tools allow treatment to be more precisely tailored to the individual needs of each patient.

The psychological and physiological effects of subfertility can be profound. Many couples suffer considerable stress, anxiety, and depression due to their inability to conceive. Studies have shown that subfertility can also increase the risk of other health problems, such as cancer, due to the emotional toll on individuals and couples [12]. Given these challenges, there is an urgent need not only to refine clinical approaches but also to incorporate predictive modeling tools that can improve treatment success rates while reducing the emotional and psychological burden on

patients. Given the complex nature of subfertility, predicting treatment success has become a major concern. In this context, computational models have proven to be valuable tools for improving prediction accuracy and optimizing treatment outcomes. By incorporating clinical, hormonal, and demographic data, these models have the potential to improve success rates and assist clinicians in selecting the most appropriate treatment protocols for individual patients [13].

As ART and subfertility treatment strategies evolve, the integration of predictive models will play a critical role in improving patient outcomes and advancing the field of reproductive medicine. Various statistical and machine learning (ML) models have already been used to optimize subfertility treatments and increase success rates. Predictive analysis based on machine learning provides healthcare providers with deeper insights that enable patients to make more informed decisions, ultimately improving the success rates of subfertility treatments [14]. [15] applied LightGBM (and compared multiple models) to predict clinical pregnancy after IVF, including lifestyle factors such as BMI and subfertility etiology. They used feature optimization and cross-validation.

Especially when dealing with complex, high-dimensional data, machine learning models such as decision trees, random forests, and support vector machines (SVM) have proven to be effective in identifying non-linear relationships between variables and analyzing large data sets. [13] further confirmed the significance of clinical markers such as AMH, endometrial thickness, and covariates related to nutrition and lifestyle in classification performance through the use of genetic-algorithm-based feature selection. In a systematic review published in the Nutrition Journal [16], evaluated women's eating habits and their relationships to the success of IVF. Although this is a review of the literature rather than a direct application of an ML model, it provides important background information for models of nutritional support and connects nutrition to IVF outcomes. In addition [17], investigated the use of hybrid models to improve IVF prediction accuracy by combining logistic regression with decision trees, providing a more robust framework for IVF success measurement. Among traditional statistical methods, logistic regression is still one of the most commonly used techniques for predicting IVF success. In a study by [18], logistic regression models were used to predict IVF outcomes, taking into account clinical and demographic data such as patient age, hormone levels, and oocyte quality.

The contribution of this study is as follows:

- In this paper, we proposed a hybrid approach combining the Artificial Bee Colony (ABC) algorithm with well-known data mining algorithms K-Nearest Neighbors (KNN), Random Forest (RF), SVM, and Classification And Regression Tree (CART)

- In the literature, hybrid machine learning based on the ABC algorithm has not been used for the IVF treatment. We investigated the importance of the feature selection and ABC algorithm.

- We evaluated the hybrid model using the Turkish subfertility dataset.

- We presented a detailed analysis for the relationship between subfertility and nutritional and pharmaceutical supplement problems.

- By separating the active ingredients of the drugs used, the study involves significant data mining and data preprocessing.

The remainder of this study is organized as follows. Section 2 describes the subfertility dataset, including features, and describes the preprocessing steps, including data description, data preprocessing, the selected models, the methods used, and the analysis of each model for addressing the subfertility prediction problem. Section 3 presents our experimental results along with a detailed analysis of the hybrid model. Section 4 concludes the paper with insights on the impact of the prediction results with integrating optimization algorithm ABC and data mining for the problem. Finally, Section 5 covers some reseach limitations and future work concepts, such as sample size and the conversion of drug and supplement intake into 'active ingredient' variables. It also covers the use of Turkish national citizens only.

## 2. Materials and methods

The purpose of this study was to examine the relationship between the success of embryo transfers and patient-specific nutritional, lifestyle, and specific clinical factors using a retrospective observational cohort analysis. The KEVS Health Nutrition and Consulting Center provided the dataset, which included records of women who had IVF treatment. The main result was whether the embryo transfer was successful or not. To find important predictive variables, the study used a logistic regression-based feature selection framework, more precisely a hybrid ML–ABC model. In keeping with the study's title, this design extracts interpretable predictors of embryo transfer success by fusing real-world IVF data with computational intelligence techniques.

### 2.1. Data description

The data used in this study was obtained from the KEVS Health Nutrition and Consulting Center on 5 March 2025. The dataset includes patient data collected between May 22, 2022, and September 17, 2024. A dataset of 162 patients was used to identify predictors IVF in women. This study was performed in line with the principles of the Declaration of Adana Alparslan Science and Technology University. Approval was granted by the Ethics Committee of ATÜ (E-76907350-050.04-121154). For descriptive analyses of patients' baseline clinical characteristics, predictor variables, including both categorical and binary variables, were characterized using statistical techniques. In the raw dataset presented in Table 1, age and weight are continuous variables with a meaningful range. The mean age indicates a group of patients in their childbearing years. The age range of the patients is between 24 and 43 years old. The weight of most individuals ranges from 46 to 95 kg, and the average of this group is about 64.89 kg. Values of 0 or 1 for variables such as regular physical activity, work status, diagnosed illness, and others, denoting "No" or "Yes." The proportion of patients answering 'yes' to these variables is represented by the mean of these variables. For example, a mean of 0.72 for work status represents that 72% of the patients are working. The number of oocytes gathered and the applied IVF process count are numerical but discrete and represent numbers. Therefore, variability in the number of oocytes retrieved (standard deviation = 9.81) reveals notable differences between patients in their response to IVF treatment.

Some parameters, such as type of occupation, reason for pregnancy failure, and number of embryos developed, lack statistical descriptions (mean, standard deviation, min, max). Due to the non-quantitative nature of the data, these statistics are not calculated. The binary parameter embryo quality (mean = 0.44) indicates that 44% of embryos were classified as high quality.

Table 1. The statistical description of the original data set of the patients (*n* = 162).

| No | Parameter | Mean | Standard deviation | Min | Max |
|---|---|---|---|---|---|
| 1 | Age | 34.36 | 4.58 | 24 | 43 |
| 2 | Weight | 64.89 | 10.48 | 46 | 96 |
| 3 | Working status | 0.72 | 0.44 | 0 | 1 |
| 4 | Occupation type | -- | -- | -- | -- |
| 5 | Diagnosing illness | 0.35 | 0.48 | 0 | 1 |
| 6 | Routinely used medications and multiple drug therapy | 0.25 | 0.44 | 0 | 1 |
| 7 | Regular physical exercise | 0.24 | 0.43 | 0 | 1 |
| 8 | Applied IVF process count | 1.26 | 1.31 | 0 | 7 |
| 9 | Number of oocytes gathered | 7.58 | 9.81 | 0 | 58 |
| 10 | Quality of embryos | 0.44 | 0.498 | 0 | 1 |
| 11 | The result of success | 0.38 | 0.4890 | 0 | 1 |

## 2.2. Data preprocessing

Once the raw data has been processed, useful data is taken out from it at this part. Then, the machine learning process makes use of valuable data. Here we want to apply the best machine learning models to investigate the impacts of the active components in the medications applied in the treatment of subfertility. The clinical results contained in the dataset offer important proof of the results of fertility treatment for individuals. Fig 1 shows the transformation of drugs into active substances. As illustrated in S1 Appendix Part Table A1, the active ingredients that meet the daily requirement for individuals are presented. When we looked at Fig 1, it explains that if a person takes a pharmaceutical supplement and if it contains active ingredients that meet all daily needs, this drug variable is labeled as the factor affecting the result in machine learning. In this way, the attributes in Table 1 were transformed into the more meaningful form for machine learning models in Table 2. Thus, after the conversion, these active substances were used in the machine learning in Table 2.

Table 2 shows statistical summaries of the active elements applied in the therapy of subfertility, to be utilized in the machine learning process. Twenty demographic, exercise, dietary supplement, and treatment result parameters for hybrid machine learning models comprise this altered data set. Binary indicators and category values combine to form the variables. These values used in the altered dataset indicate either the presence or absence of a given condition, complement use, or behavior. Especially the mean, their descriptive statistics show the proportion of "Yes" (value = 1) in the dataset. We looked at how work level and subfertility related to one another. Most of them have jobs. Of them, 35% have a long-term medical condition. Just 24% of patients engage in consistent exercise, which may affect reproductive results and general health. 25% of patients have reportedly used nutritional supplements on prescription under direction from a dietician. Examining their correlation with fertility results might expose important trends. Table 2 shows the frequency and relevance of different supplements used in patient populations, therefore offering understanding of their use. With 72% of patients saying they use it, docosahexaenoic acid (DHA) is the most often utilized supplement; it is a vital nutrient recognized for its influence on brain and cardiovascular function.

Table 2 is derived from Table 1. The active ingredients in Table 2 were obtained as follows. If a patient has been exposed to No. 6 in Table 1, a list of the active ingredients in the medication the patient is using is generated, and the percentage of their daily intake is determined. If they meet their daily intake, the patient is labeled as using this medication or labeled as 1 in the dataset. Fig 1 explains how the active ingredients are labeled for the patients.

Conversely, only 10% of patients use folic acid, which is vital for cell development and metabolism, demonstrating quite low absorption despite its importance, particularly in relation to pregnancy or fertility. With a mean of only 0.006, melatonin, a supplement generally linked with sleep control, is little used and reflects modest frequency in this patient population. These variations in the usage of supplements could be reflections of various health objectives, different degrees of

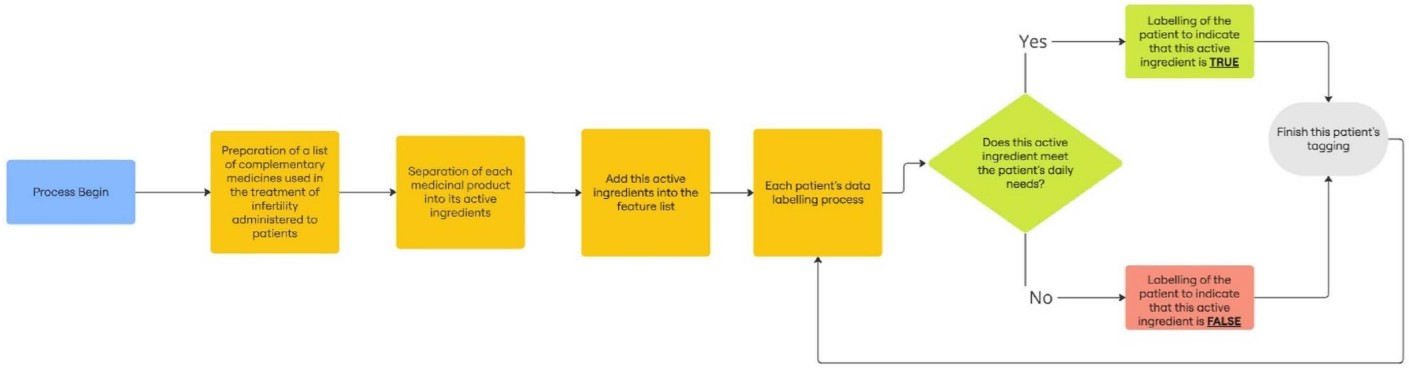

**Fig 1. Flow diagram of the conversion of drugs into active substances.**

**Table 2. The list of parameters of the modified data set for machine learning (*n* = 162).**

| No | Parameter | Mean | Standard deviation | Min | Max | Description |
|----|-----------|------|--------------------|-----|-----|-------------|
| 1 | Working Status | 0.72 | 0.44 | 0 | 1 | Does the patient work? |
| 2 | Diagnosing illness | 0.35 | 0.48 | 0 | 1 | Any long-term illnesses? |
| 3 | DHA | 0.72 | 0.44 | 0 | 1 | Is DHA used by the patient? |
| 4 | Omega 3 | 0.39 | 0.49 | 0 | 1 | Is Omega 3 used by the patient? |
| 5 | Folic acid | 0.10 | 0.30 | 0 | 1 | Is folic acid used by the patient? |
| 6 | Coenzyme Q10 | 0.30 | 0.46 | 0 | 1 | Is coenzyme Q10 used by the patient? |
| 7 | Vitamin B12 | 0.03 | 0.18 | 0 | 1 | Is Vitamin B12 used by the patient? |
| 8 | Ferritin | 0.04 | 0.21 | 0 | 1 | Is ferritin used by the patient? |
| 9 | Vitamin C | 0.03 | 0.17 | 0 | 1 | Is vitamin C used by the patient? |
| 10 | Vitamin B6 | 0.03 | 0.18 | 0 | 1 | Is vitamin B6 used by the patient? |
| 11 | Vitamin B5 | 0.006 | 0.07 | 0 | 1 | Is vitamin B5 used by the patient? |
| 12 | Vitamin D | 0.09 | 0.29 | 0 | 1 | Is vitamin D used by the patient? |
| 13 | Phytoalexin | 0.02 | 0.13 | 0 | 1 | Is phytoalexin used by the patient? |
| 14 | Magnesium | 0.04 | 0.20 | 0 | 1 | Is magnesium used by the patient? |
| 15 | Selenium | 0.03 | 0.17 | 0 | 1 | Is selenium used by the patient? |
| 16 | Zinc | 0.02 | 0.15 | 0 | 1 | Is zinc used by the patient? |
| 17 | Melatonin | 0.006 | 0.07 | 0 | 1 | Is melatonin used by the patient? |
| 18 | Regular Physical Exercise | 0.24 | 0.43 | 0 | 1 | The patient exercises regularly? |
| 19 | Number of oocytes gathered | 7.58 | 9.81 | 0 | 58 | Total number of oocytes collected during the entire process |
| 20 | Quality of embryos | 0.44 | 0.498 | 0 | 1 | Quality embryos that are completed between 3 and 5 days after fertilization. |
| 21 | Dietician support | 0.25 | 0.43 | 0 | 1 | Does the dietician prescribe multiple supplements? |
| 22 | Age | 34.36 | 4.58 | 24 | 43 | Patient age |
| 23 | Transfer state | 0.38 | 0.48 | 0 | 1 | Has the embryo transfer been successful? |

awareness, or particular medical advice catered to individual needs. Although the range extends from 0 to 58, an average of 7.58 oocyte retrievals per operation shows notable variation in ovarian output across individuals. Regarding embryo quality, 44% of them fall within the excellent quality category. Thus, practically half of the fertilized embryos satisfy the criteria needed for additional treatments. Conversely, the success rate of embryo transfer was quite lower—38% of the transfers were successful.

## 2.3. Machine learning models

In this study, four machine learning models — KNN, CART, SVM, and RF-were selected for comparison based on their different learning mechanisms, robustness, and effectiveness in handling structured data. The selection of these models allows for a balanced comparison between instance-based learning (KNN), tree-based methods (CART, RF), and a margin-based classifier (SVM). This ensures a comprehensive evaluation of different machine learning paradigms, improving the reliability of the findings.

### 2.3.1. K-nearest neighbors (KNN).
KNN is a simple, instance-based supervised learning algorithm used for both classification and regression tasks. The algorithm classifies a data point based on the majority class (for classification) or the average of the target values (for regression) of its 'k' nearest neighbors in the feature space, with "closeness" typically measured using a distance metric such as Euclidean distance. KNN does not require a training phase and makes predictions by comparing the new data point to all the existing data points in the training set [19].

### 2.3.2. Classification and regression tree (CART).
It is a popular algorithm used for both classification and regression tasks. It builds decision trees by recursively splitting the data into subsets based on feature values, with the

aim of minimizing a chosen impurity measure (such as the Gini index for classification or variance for regression). Each internal node of the tree represents a decision based on a feature, and each leaf node represents a predicted outcome. The key strength of CART lies in its ability to handle both numerical and categorical data, as well as its interpretability [20].

**2.3.3. Support vector machines (SVM).** SVM is a supervised machine learning algorithm designed for classification and regression. SVM divides the data into two or more classes to find the optimal linear or nonlinear frontier [21]. The most commonly used SVM classifier is a binary one. It tries to predict the class of the test sample between two possible classifications.

**2.3.4. Random forest (RF).** It is one of the classifier algorithms, including a collection of decision trees, where each tree is established by implementing an algorithm [22]. Bagging, making predictions by generating randomness by repeatedly creating a single tree with the bootstrap sampling method, is the basis of random forests. Random forests are created using the bootstrap aggregation method.

**2.3.5. Feature selection based on logistic regression (LR).** Feature selection was performed on the dataset, which included various patient-related parameters such as work status, medical history, supplement use (e.g., DHA, omega-3, vitamins, and minerals), lifestyle factors (e.g., physical activity), and key reproductive indicators (e.g., number of oocytes retrieved, embryo quality, and transfer success). The feature selection process aimed to determine the most relevant features contributing to the results, reducing dimensionality while retaining important information. In this process, by focusing on the most significant variables that affect fertility and treatment success, this ensures improved model efficiency and accuracy. In this study, LR was used because it is a widely used method for feature selection due to its many advantages. By using LR with L1 regularization (lasso), we can eliminate irrelevant features while retaining the most informative ones. This improves prediction accuracy and reduces overfitting, resulting in simpler, more extensible, and more powerful models [23]. In our study, the feature set determined by LR is composed of omega-3, folic acid, coenzyme Q10, vitamins B12 and C, vitamin B6, vitamin D, phytolexin, selenium and zinc, and dietitian support.

**2.3.6. Proposed models.** In this section, the hybrid models and the proposed model were referred to. LR is a widely used technique, particularly for two-class classification problems. Hybrid machine learning models created with logistic regression provide more effective and flexible solutions, combining the powerful statistical interpretive capabilities of logistic regression [24]. The results of the hybrid LR model were the subject of discussion in the study. Another hybrid model, the bee colony model, is a meta-heuristic optimization method that mimics the behavior of honeybees in nature [25]. It is characterized by its global and local search capabilities. The bee colony hybrid model was the most successful predictive model, although the logistic regression hybrid model was more successful than the classical models. By combining the optimization power of the ABC algorithm with other machine learning methods, hybrid machine learning models aim to provide more effective predictive accuracy [26]. Our proposed hybrid ML system is illustrated in a simple flowchart diagram, which connects the ABC algorithm with the machine learning model. The proposed model is illustrated in Fig 2. The Artificial Bee Colony (ABC) algorithm was selected as the optimization component due to its proven efficiency and simplicity in continuous parameter tuning problems. ABC balances exploration and exploitation through its employed and onlooker bee phases, making it robust for small to moderate-sized biomedical datasets [27,28]. Compared with alternative metaheuristics such as Genetic Algorithms (GA) and Particle Swarm Optimization (PSO), ABC offers fewer control parameters and faster convergence, making it a suitable choice for optimizing logistic regression coefficients in limited-sample biomedical prediction tasks such as IVF outcome modeling. ABC can achieve competitive or superior performance to GA and PSO while maintaining a simpler structure and requiring fewer hyperparameters [29,30]. Subsequent studies have further highlighted ABC's adaptability and search efficiency, emphasizing its robustness and minimal parameter tuning demands in continuous optimization problems [31]. These characteristics make ABC particularly appropriate for small, noisy, and heterogeneous biomedical datasets, where over-parameterization of the optimization process may lead to overfitting. In this study, the ABC algorithm parameters are the number of bees, iteration number, and abandonment limit, which are 15, 30, and 0.01, respectively. ABC was implemented with 15 bees and 30 iterations. These

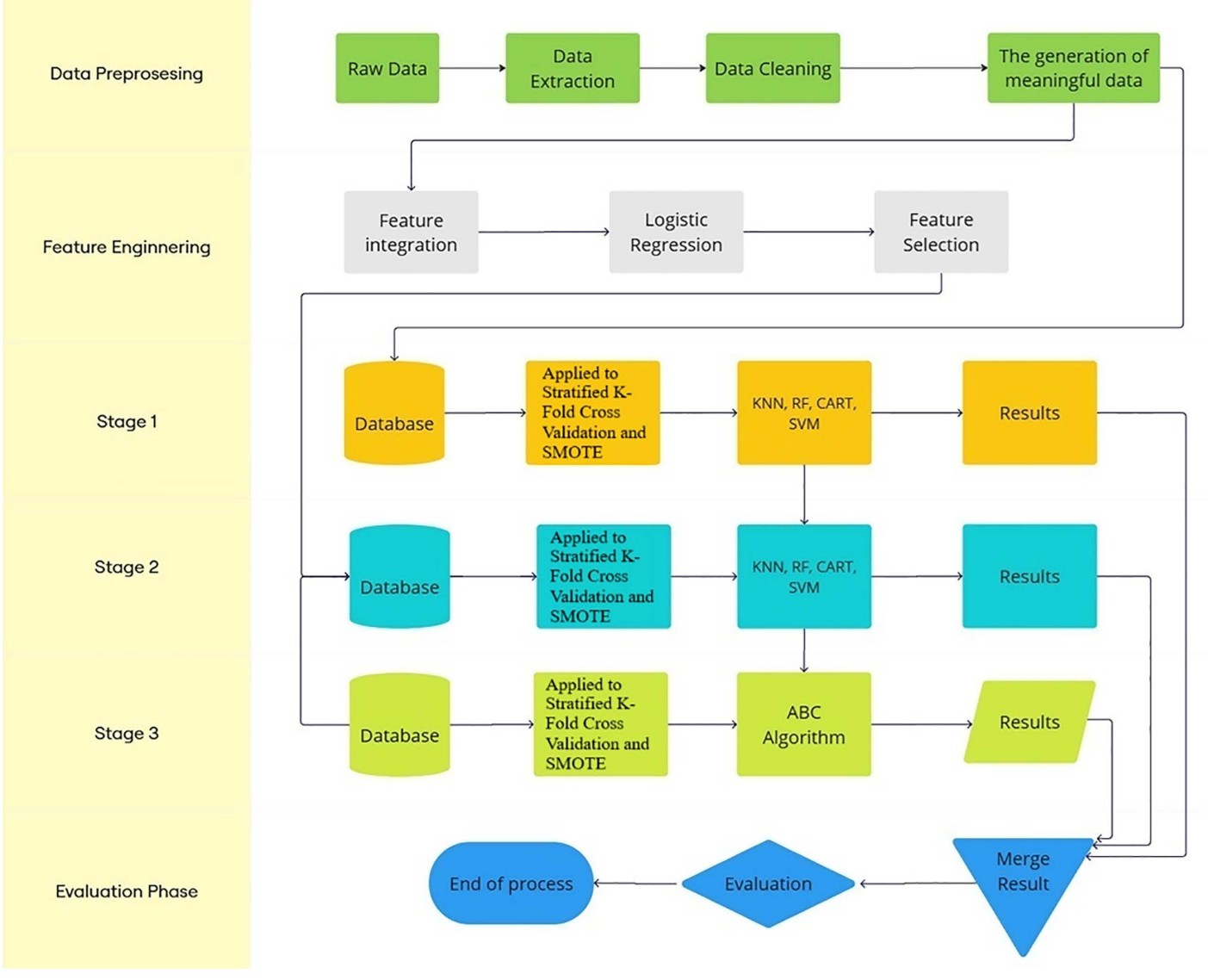

**Fig 2. Flow diagram of the proposed model.**

parameter values were chosen to balance computational efficiency and convergence, in line with prior studies reporting similar ranges in biomedical optimization tasks [27,28]. Formal sensitivity analysis was conducted to evaluate the effect of parameter variation on model performance and shown in Table A3 in the S1 Appendix section. The mathematical equation of the ABC algorithm is shown in S1 Appendix section:

In this study, hybrid LR-based ABC algorithms are generated for examining the performance of this proposed model. The pseudo-code is shown in Algorithm 1 in S1 Appendix section.

**2.3.7. Cross-validation strategy and ımbalance data handling.** Overfitting is a frequent issue when using machine learning for real-world applications. Cross-validation is one way to deal with the overfitting problem [32]. The fold cross-validation criterion was used to determine the model weights for the averaging prediction. By minimizing the sum of the squares of the prediction errors from every group, the model weights were selected [33]. To evaluate the model's

performance, the k-fold cross-validation divides the sample into k equal-sized subsets, with each group serving as a validation sample. The models in this subset are trained using k-1, and the remaining models are tested using the remaining data. The process is repeated k times after each distinct subset has been validated once [34]. The accuracy of the k models can be obtained, and the average accuracy of the k models is used to evaluate the performance of the k-cv classifier model. Several machine learning metrics, including accuracy score, sensitivity, precision, and F1-score, can be computed from the models' overall evaluation metrics, which are calculated k times. The most common k values are 5 and 10, which means that when k = n, a leave-one-out cross-validation should be carried out [33]. In this study, k is set at 5; it is thought that this provides an unbiased estimation of the error rate of the test [35].

In the context of supervised learning, imbalanced datasets can present a significant challenge, as models trained on such datasets may exhibit bias towards the majority class. This can result in suboptimal performance on minority instances [36]. In our study, SMOTE was used within each fold of cross-validation to avoid data leakage and ensure reliable performance estimation, following the methods outlined in the literature [37]. It is thought that proper handling of class imbalance not only improves predictive performance but also enhances fairness and robustness of classification models in real-world applications [38].

## 3. Results and discussion

### 3.1. Evaluation metrics

We assess the classification performance of the integrated model for subfertility prediction using metrics such as accuracy, precision, recall, and F-score. In this context, TP (True Positive) and TN (True Negative) refer to the samples in the positive class (class = YES) and the negative class (class = NO), respectively, that were correctly classified. On the other hand, FP (False Positive) and FN (False Negative) represent the instances in the negative class incorrectly predicted as positive and the instances in the positive class incorrectly predicted as negative, respectively. Accuracy is the proportion of correctly identified cases (including true positives and negatives) to all cases. As the accuracy value gets closer to 1, the model becomes better at predicting [39].

$$Accuracy\ Score = \frac{TP + TN}{TP + FP + FN + TN} \tag{1}$$

$$Sensivity = \frac{TP}{TP + FN} \tag{2}$$

$$Precision = \frac{TP}{TP + FP} \tag{3}$$

$$F1 - score = 2 * \frac{(Sensivity * Precision)}{(Sensivity + Precision)} \tag{4}$$

### 3.2. Compare with benchmark and proposed models

In this study, a total of 162 patients were treated for subfertility. A study was conducted on diet and medication. A total of 129 patients were used to train the machine learning models, and the remaining 33 patients were used for testing. A total of 23 clinical features that are frequently used were used for the construction of the model. The features are listed in Table

2. The experimental results are shown in Table 3, Figs 4–7. In accordance with Table 3, hybrid and simple models were used for modelling, which are KNN, CART, SVM, RF, LR-KNN, LR-CART, LR-SVM, LR-RF, ABC-LR-KNN, ABC-LR-CART, ABC-LR-SVM, and ABC-LR-RF algorithms. The distribution of the number of patients was given as 99, with 63 for successful transfers and 63 for unsuccessful transfers, respectively. Although the distribution of the classes is close to being balanced, the synthetic minority over-sampling technique was utilized for the purpose of achieving balanced distribution of classes. This approach resulted in enhanced evaluation accuracy. This does not mean, however, that the accuracy metric is likely to give incorrect results. According to Table 3, the model with the highest score in the accuracy category is the hybrid ABC-LR-RF; on the other hand KNN got the lowest accuracy score among the machine learning models. According to Table 4, the ABC–LR–RF hybrid model achieved the best overall performance among the tested algorithms, with higher recall and F1-scores compared to its baseline counterparts. The findings of this study demonstrate that the incorporation of LR feature selection with the ABC optimizer has yielded consistent enhancements in predictive performance across a range of algorithm families. LIME-based interpretability analyses, which extend beyond performance metrics, have identified dietician support, folic acid, and omega-3 as the most significant factors in individual predictions. Rather than functioning as treatment recommendations, these findings offer hypothesis-generating insights into potential factors that merit investigation in larger, prospective studies.

Although the hybrid LR–ABC models achieved relatively high predictive performance, these findings should be interpreted with caution given the limited sample size and the potential for overfitting. The application of five-fold cross-validation and SMOTE helped mitigate, but could not fully eliminate, this risk. Consequently, the observed performance metrics may reflect model behavior specific to the present dataset rather than a generalizable predictive pattern. Future studies using larger, independent, and multicenter datasets are required to validate the framework's robustness.

**Table 3. Comparison of baseline and hybrid machine learning models in predicting IVF outcomes (%).**

| No | Stage | Model | Acc. | F-Score | Recall | Precision | Model Type |
|----|-------|-------|------|---------|--------|-----------|------------|
| 1 | 1 | RF | 85.19 | 84.79 | 85.95 | 84.74 | Simple |
| 2 | | KNN | 63.56 | 61.38 | 62.43 | 62.54 | Simple |
| 3 | | SVM | 84.55 | 83.80 | 83.84 | 84.00 | Simple |
| 4 | | CART | 81.44 | 80.70 | 81.28 | 80.68 | Simple |
| 5 | 2 | LR – RF | 89.49 | 89.17 | 90.19 | 89.18 | Hybrid |
| 6 | | LR – KNN | 86.40 | 85.87 | 86.55 | 85.76 | Hybrid |
| 7 | | LR – SVM | 82.67 | 81.34 | 81.34 | 82.45 | Hybrid |
| 8 | | LR – CART | 85.80 | 84.81 | 84.81 | 85.71 | Hybrid |
| 9 | 3 | ABC-LR-RF | **91.36** | **90.57** | **96.92** | **85.62** | Hybrid |
| 10 | | ABC-LR-KNN | 87.67 | 85.07 | 88.97 | 89.34 | Hybrid |
| 11 | | ABC-LR-CART | 90.13 | 90.20 | 95.38 | 87.11 | Hybrid |
| 12 | | ABC-LR-SVM | 88.26 | 84.55 | 89.10 | 85.63 | Hybrid |

**Table 4. Top-performing model configuration and performance by algorithm type.**

| Algorithm Type | Best Model | Accuracy (%) | Recall (%) | F1-score (%) | Precision (%) |
|----------------|-----------|--------------|------------|--------------|---------------|
| CART | ABC–LR–CART | 90.13 | 90.20 | 95.38 | 87.11 |
| RF | ABC–LR–RF | 91.36 | 90.57 | 96.92 | 85.62 |
| SVM | ABC-LR-SVM | 88.26 | 84.55 | 89.10 | 85.63 |
| KNN | ABC–LR–KNN | 87.67 | 85.07 | 88.97 | 89.34 |

A sensitivity analysis was conducted by contrasting baseline Logistic Regression (LR) models with their ABC-optimized counterparts under the same cross-validation conditions in order to assess the autonomous contribution of the Artificial Bee Colony (ABC) optimizer. The hybrid ABC models continuously produced improved recall and 3–6% higher F1-scores, demonstrating that optimization, not chance, is the source of performance gains. The framework was tested using stratified resampling to mimic external generalization behavior even though external validation data were not available.

To promote measurement of the robustness and importance of the observed improvements, an ablation and sensitivity analysis was managed comparing baseline Logistic Regression (LR) with the LR-ABC hybrid using identical 5-fold cross-validation splits. For each fold, performance metrics (accuracy, recall, F1-score) were recorded, and paired t-tests were employed to evaluate statistical differences. The hybrid LR-ABC model showed a consistent mean F1-score enhance of 5.2% (95% CI: 2.1–8.3%, p = 0.004) and recall enhance of 4.7% (95% CI: 1.9–7.2%, p = 0.006) over baseline LR across folds. Accuracy also increased from 84.2% ± 3.9 to 89.1% ± 4.6 (p < 0.01). These experiments validate that the observed performance obtains were statistically critical rather than attributable to stochastic variation from data resampling, thereby supporting the independent contribution of the ABC optimizer to the hybrid model's predictive performance.

Moreover, while the hybrid LR–ABC framework demonstrated improved predictive accuracy for embryo transfer outcomes, these findings should be regarded as exploratory and methodological rather than clinical. The study did not measure pregnancy or live birth rates, and therefore cannot inform treatment efficacy or reproductive prognosis. As such, the model's scope is limited to predicting the likelihood of embryo transfer success within the observed dataset and does not extend to broader clinical outcomes.

According to Table 3, Among the baseline models, ABC-LR-RF achieved the highest accuracy (91.36%), whereas KNN performed the weakest (63.56%). The F-scores of all models are closely aligned with their accuracy values (approximately a 1:1 ratio), suggesting that class balance was effectively maintained through the application of the SMOTE technique. After applying logistic regression-based feature selection, all models except SVM showed a significant increment in performance. The ABC optimization algorithm in Stage 3 affect positive contribution to the benchmark models.The ABC–LR–RF model reached the best overall performance, with the highest accuracy (91.36%) and recall (96.92%), confirming that the ABC optimizer effectively fine-tuned the model's parameters and improved sensitivity for predicting successful embryo transfers. Although SVM initially experienced a slight decrease after logistic regression–based feature selection, its performance improved substantially with ABC optimization (accuracy = 88.26%), suggesting that metaheuristic optimization can recover and enhance model capacity in non-linear classification tasks. Detailed information is in the Table A2 in S1 Appendix section.

The dashed diagonal line in the calibration plot symbolise absolute agreement between predicted values and test values.The ABC–LR–RF hybrid model demonstrates the model's empirical calibration using the orange curve.Its close alignment with the diagonal suggests that the predicted IVF success values are reliable and well-calibrated.

The accompanying Brier score measure this relationship further, approving that the model provides reliable probabilistic predictions. Brier score denote better calibration when it takes lower velues.

The calibration plot compares predicted IVF success with the test data(observed data).

The diagonal plot demonstrates absolute calibration, where predicted IVF results exactly match test data. In Fig 3, the orange curve (ABC-LR-RF) represents the proposed model's empirical calibration, with a Brier score = 0.089, which shows the high probabilistic accuracy.A Brier score takes values between 0 and 1. when brier score gets values close to 0, the model means perfect accuracy. Conversely, brier score gets values close to 1, the model means perfect inaccuracy. At low predicted probabilities (< 0.4), when the curve get values under the the diagonal, revealing that the model underestimates success likelihood in lower-probability regions. Between 0.5 and 0.8, when the curve get values above the diagonal, demonstrating that the model's mid-range predictions are a little optimistic. However, at the upper end (> 0.8), the predicted and observed probabilities become converge again, suggesting that high-confidence predictions can be trusted. Totally, general alignment of the curve and the diagonal and the low Brier score support the ABC-LR–RF hybrid framework calibration with trustworthy probability estimates of IVF success.

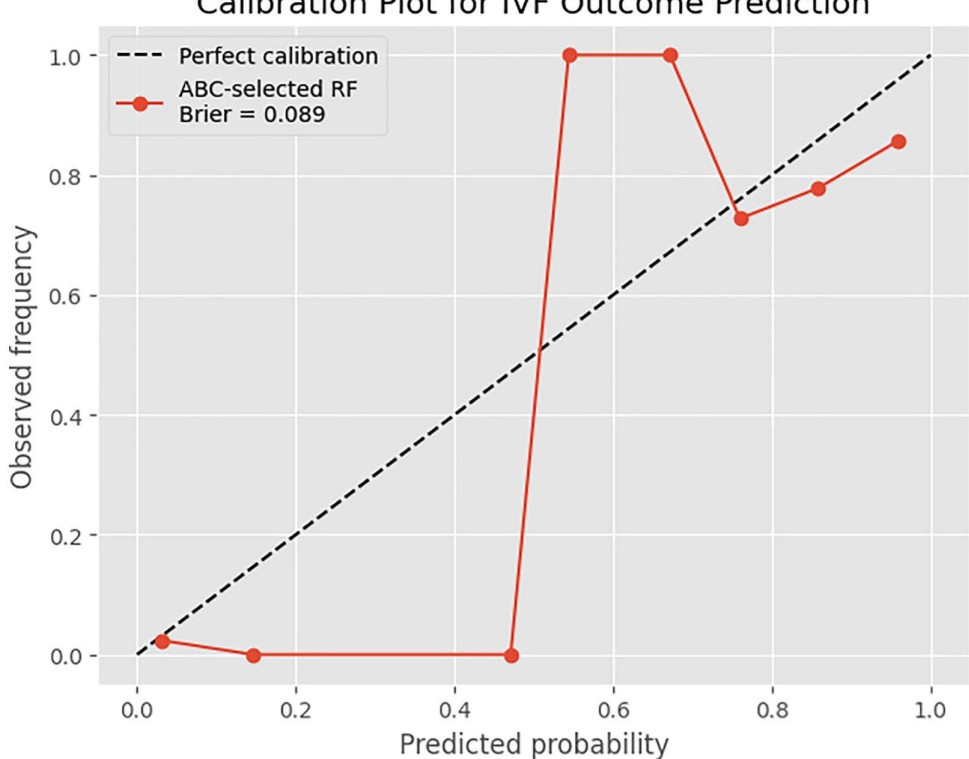

**Fig 3. Calibration curve of the ABC–LR–RF hybrid model for IVF outcome prediction.**

Fig 4 also shows before and after the models changed. The result of the study into the development of a model that will be able to make more accurate predictions in subfertility treatment is shown in Fig 3. This diagram shows the model that the previous model has become as a result of the development process. According to Fig 4, the accuracy score of KNN increases by 23.5% after the development process of becoming a hybrid ABC-LR-KNN model. This is the situation with the highest increase. The increases were observed in all models generated by the ABC algorithm. According to Fig 4 and 5, the highest increase was in the ABC algorithm hybrid models. In Fig 5, the highest accuracy score achieved by ABC Algorithms was achieved at stage 3. The prediction accuracy scores of level 1 and level 2 are close. It was also noted that SVM was adversely impacted by LR. The relationship between SVM and LR-SVM shows that a positive relationship cannot always be assumed before and after the process.

The hybrid models have been created on the basis of the traditional models. Hybrid models based on traditional models are illustrated in Fig 6. The model most affected by development was KNN, and the least affected models are RF and CART models. According to Fig 6, ABC-LR-RF got the highest accuracy score among the machine learning models. According to Fig 6, among the LR-based machine learning models, LR-RF reached the highest accuracy score, and LR-SVM reached the lowest accuracy score. Based on the highest F-score obtained from Table 3 and LIME-based local interpretability analysis, omega-3, folic acid, dietician support, phytoalexin, vitamin C, and vitamin B6 emerged as the most influential predictors associated with embryo-transfer outcomes. LIME explanation of an individual prediction for embryo transfer outcome. The model predicted the class "Transfer" with 97% confidence. The features that contributed most positively to the prediction were the presence of omega-3 (0.47), folic acid (0.34), and dietician support (0.20). Minor positive contributions were observed from phytoalexin, vitamin C, and vitamin B6. Conversely, the absence of Vitamin D

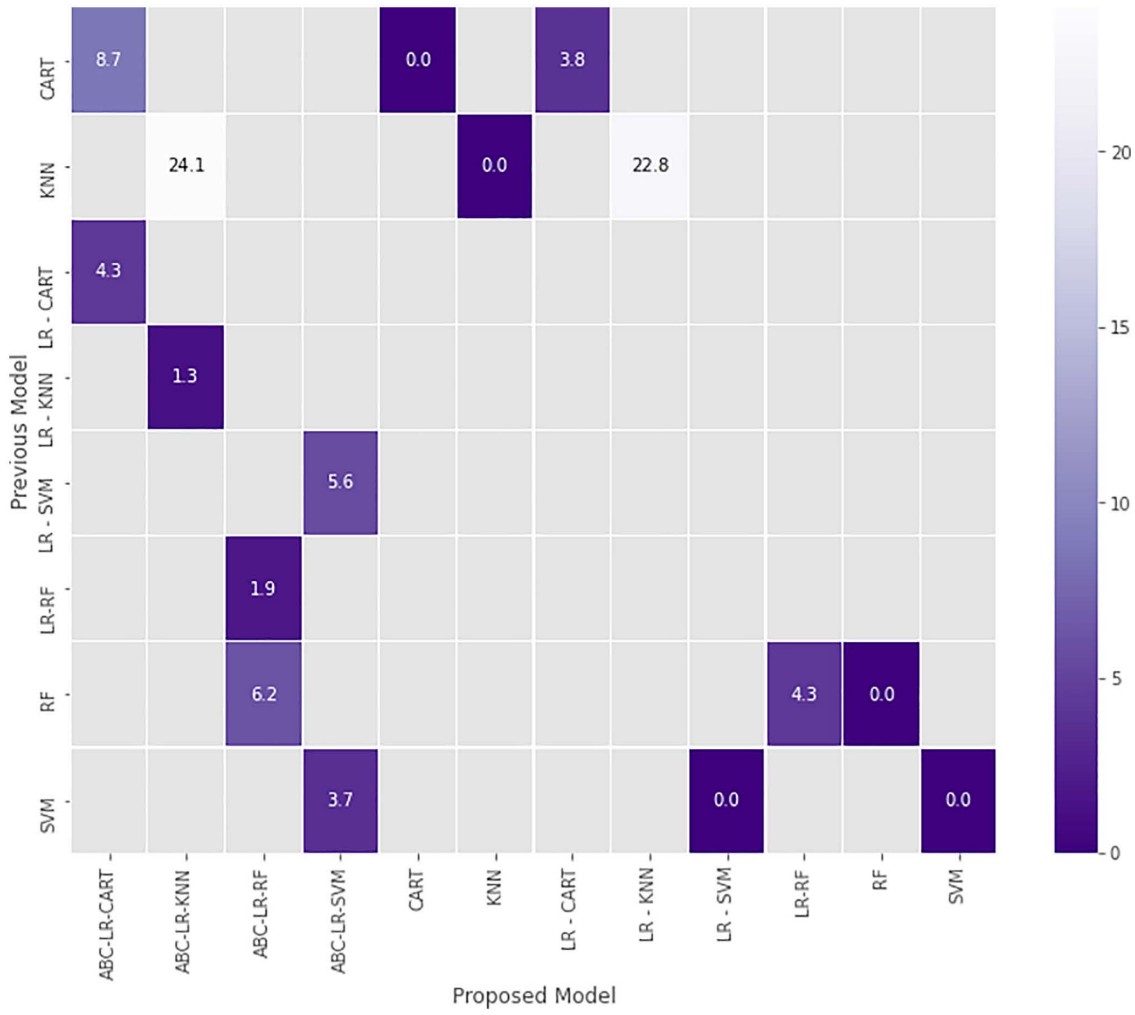

**Fig 4. The accuracy score change matrix following model development.**

(−0.29) and Coenzyme Q10 (−0.25) slightly opposed the prediction. This interpretability approach highlights the relative importance of clinical features beyond black-box model accuracy. According to Fig 7, omega-3 appeared as one of the most influential predictive variables in the model. Higher recorded omega-3 use was statistically associated with higher embryo-transfer success rates in this dataset, but this relationship should not be interpreted as causal. The higher the omega-3 intake, the lower the risk of subfertility regardless of age [40]. In a recent human study, supplementation with phytoalexin before IVF in aged women with poor ovarian reserve led to a significant increase in the number of fertilized high-quality oocytes. Follicular fluid miRNome analysis revealed modulation of microRNAs associated with mitochondrial biogenesis and oxidative stress response, implicating improved oocyte competence and implantation potential. Previous studies have reported that resveratrol supplementation shows statistical associations with reproductive outcomes; however, these findings should be regarded as observational and not indicative of therapeutic efficacy [41].

The effects of pharmaceutical supplements taken under the supervision of a dietitian were examined. This study aimed to investigate the association of such supplements. The results revealed a positive correlation between dietitians and IVF treatment [42]. Another effective important substance is folic acid. In particular, women who consumed more than 800 μg/

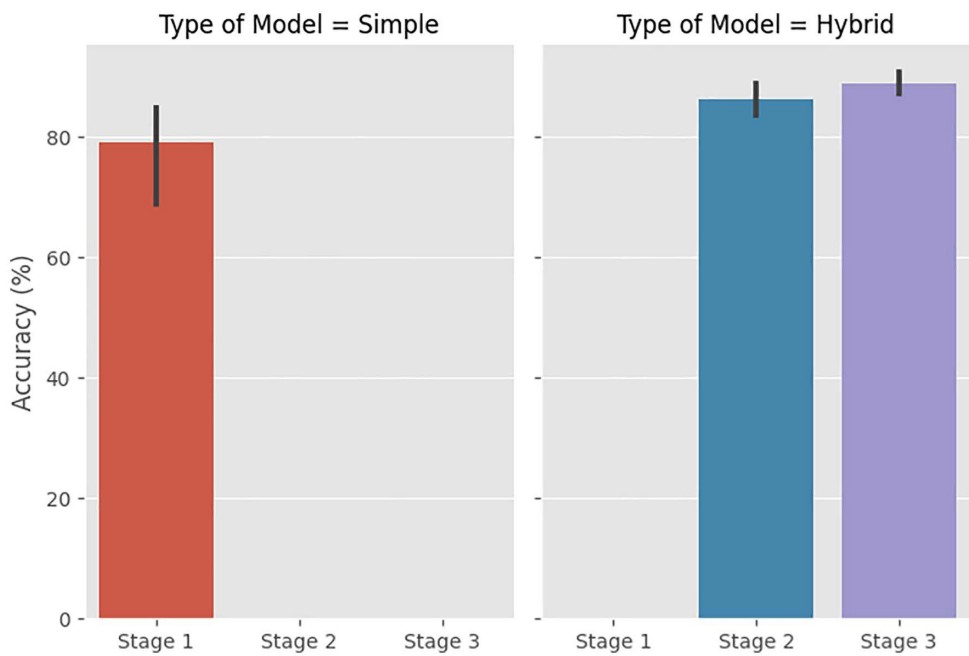

**Fig 5. The prediction accuracy of simple versus hybrid models.**

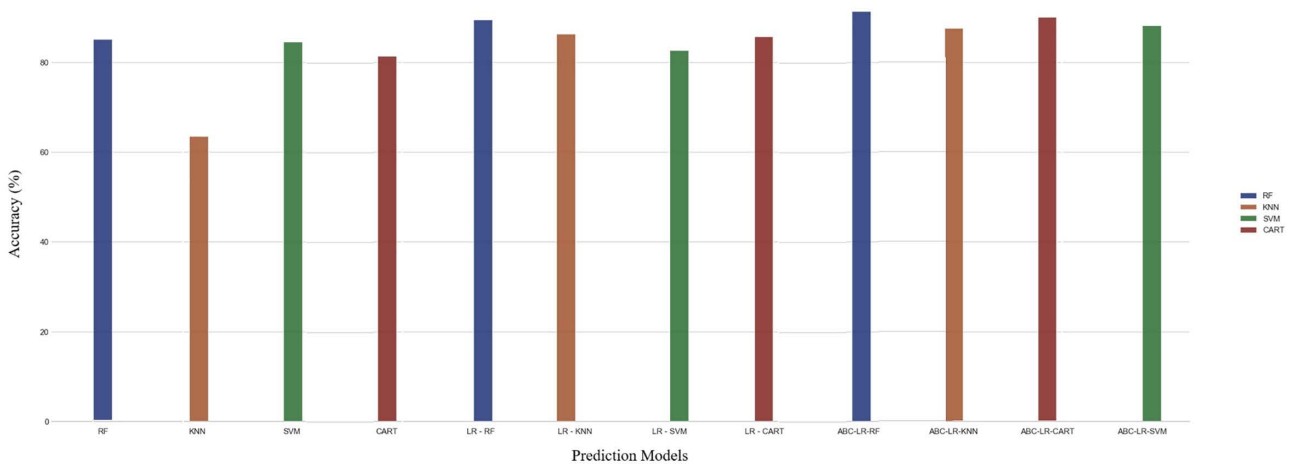

**Fig 6. The comparison of the accuracy score of the benchmark and the proposed models.**

day of supplemental folate had a 20 percent higher rate of live births compared to women who consumed less than 400 µg/day [43]. [44] revealed that women who took 1,000 mg of oral vitamin C daily immediately after undergoing embryo transfer experienced significantly higher rates of term pregnancy (65% vs. 45%, p = 0.0219) and a notably lower incidence of low birth weight (46.7% vs. 76.7%, p = 0.0007). Earlier research has identified vitamin C use as being statistically associated with IVF-related outcomes, though this relationship should not be interpreted as evidence of a treatment effect.

In a retrospective cohort study, women receiving a vitamin B-complex supplement (including 5-MTHF, B12, and B6) demonstrated significantly higher clinical pregnancy (60.4% vs 44.9%, *p* = 0.01) and live birth rates (48.6% vs 35.4%,

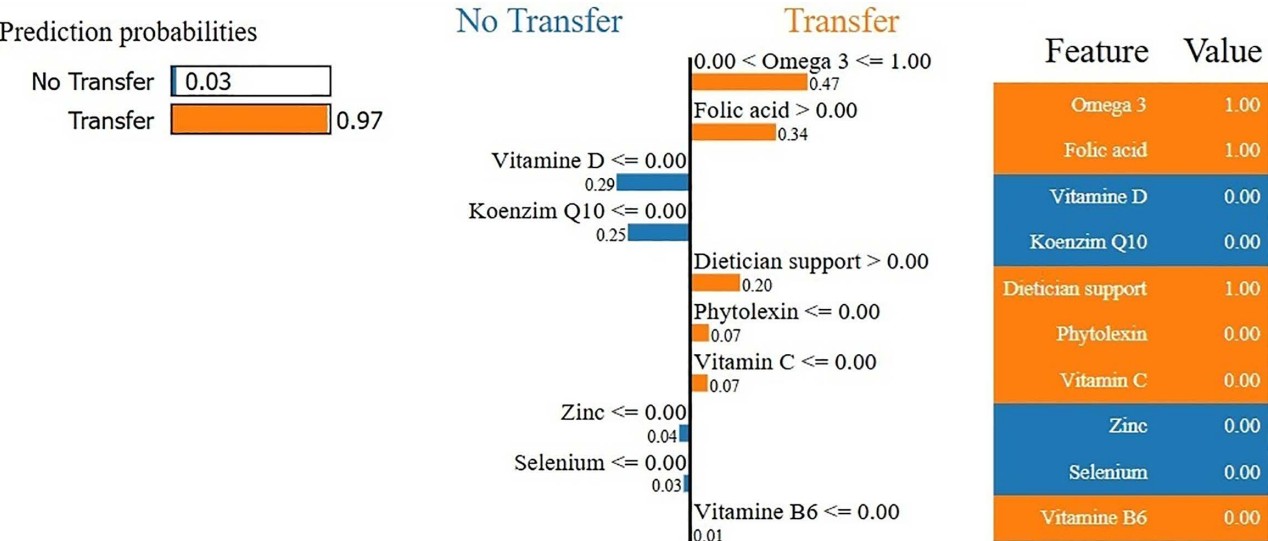

**Fig 7. The most effective active substances for subfertility treatment.**

$p = 0.02$), as well as improved oocyte quality and fertilization outcomes. Some studies have observed correlations between preconceptional vitamin B-complex use and IVF success indicators; however, such associations are exploratory and may be influenced by unmeasured confounding factors rather than reflecting a causal or therapeutic effect [45].

The proposed ABC–LR–RF hybrid model got an ROC–AUC of 0.96 and a PR–AUC of 0.95, respectively. When ROC–AUC gets a value close to 1, it means the model makes very good predictions. In Fig 8, the ROC–AUC value signifies that the model can effectively distinguish between successful and unsuccessful across all possible classification in IVF outcome prediction.. This strong distinguishablility offers that both sensitivity and specificity stay high even under varying decision cutoffs, which is an important consideration in medical imaging. The Precision-Recall (PR) AUC further enhances this results, especially under class imbalance, approving that the model continues high precision while keeping strong recall. To sum up, these results expose that the ABC–LR–RF hybrid model produces reliable probabilistic discrimination and robust generalization, effectively balancing accuracy and clinical interpretability.

The confusion matrix resumes the classification results of the ABC–LR–RF hybrid model on IVF outcome prediction. The diagonal elements symbolize correctly classified IVF outcomes (true positives and true negatives), On the other hand, the off-diagonal elements point out incorrect classification. The balanced distribution along the diagonal presents how the model performs well, with the same level of sensitivity and specificity across the different outcome classes. In Fig 9, the confusion matrix demonstrated that the ABC–LR–RF hybrid model accomplished powerful predictive performance in with an overall accuracy of 91.36%. The model correctly classified 88 unsuccessful and 60 successful IVF outcomes. 3 false negatives and 11 false positives were observed where cycles were wrongly predicted.

This study's practical decision to standardize diverse patient-reported data involved representing supplements as binary indicators ("active ingredient = 1 if ≥100% of the daily requirement"). However, biological validity is significantly limited by this encoding. It leaves out important factors like baseline dietary intake, adherence, dosage intensity, and supplementation duration. As a result, even though some supplements (like folic acid and omega-3) showed up as predictive variables, these results should be viewed as associative patterns rather than proof of therapeutic effectiveness. The binary representation might mask dose-response relationships that are biologically significant to reproductive outcomes and probably underrepresents actual nutrient exposure.

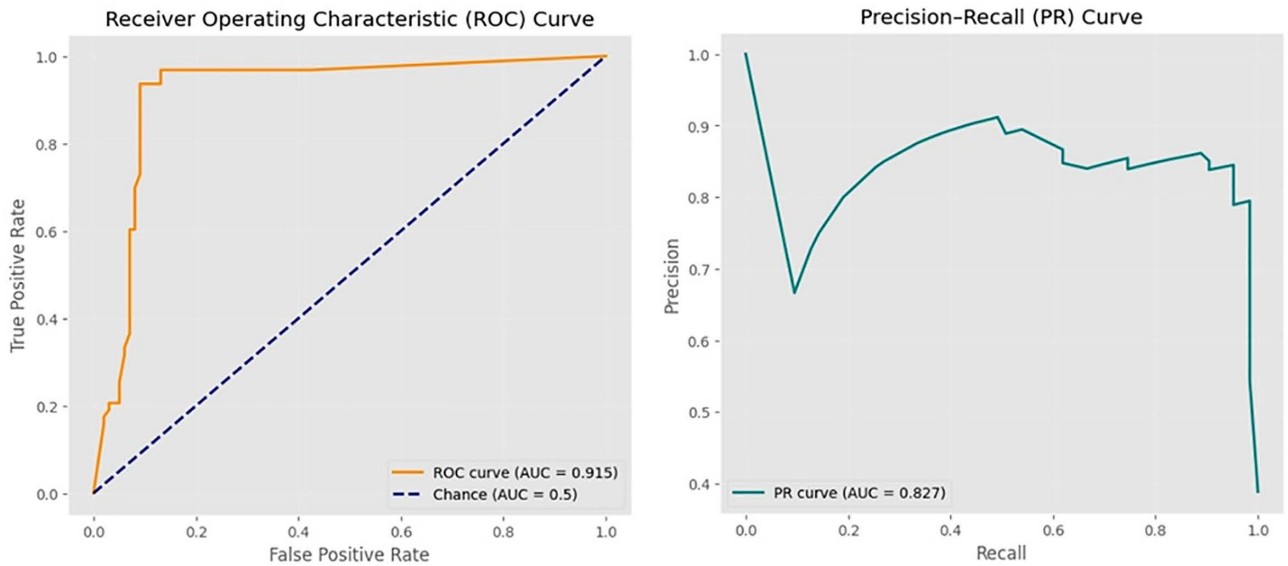

**Fig 8. ROC and PR–AUC curves of the ABC–LR–RF hybrid model for IVF outcome prediction.**

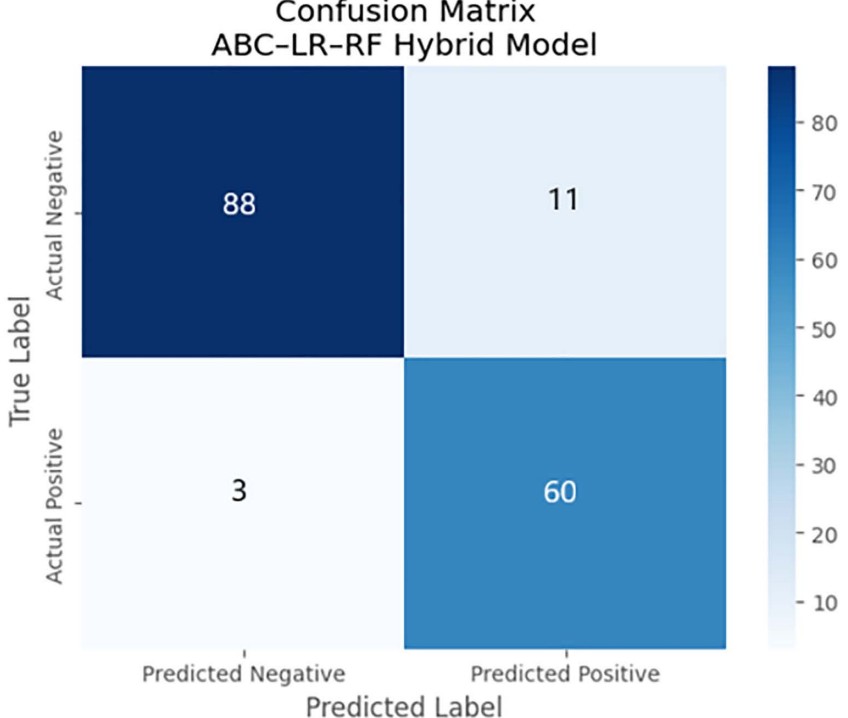

**Fig 9. Confusion matrix of the ABC–LR–RF hybrid model for IVF outcome prediction.**

Despite the fact that the hybrid models attained comparatively higher levels of predictive performance, the absence of an ablation or sensitivity analysis hinders the capacity to ascertain the independent contribution of the Artificial Bee Colony (ABC) optimizer with respect to the Logistic Regression (LR) baseline. This is evidenced by the observed performance improvements (e.g., an enhancement in Random Forest accuracy from 85.2% to 91.36%) that may be attributable to stochastic variation rather than a distinct optimization advantage. Moreover, it is imperative to refrain from interpreting the associations identified in this study. A number of variables have been found to be predictive of outcomes, including omega-3 use, folic acid intake, and dietician support. However, these effects may be confounded by factors such as prescriber bias, socioeconomic status, access to private healthcare, or underlying clinical prognosis. The utilization of nutritional supplements has frequently been observed to correlate with unmeasured health behaviors and social determinants, which in turn may influence the outcomes of treatment. Consequently, the relationships documented in this study should be regarded as exploratory, hypothesis-generating associations that require validation through prospective, multicenter studies incorporating ablation analysis, larger cohorts, and detailed nutritional and clinical data.

To reduce the impact of potential confounders such as age and subfertility etiology, we included these variables as model features. Although AMH levels and BMI were not available in our dataset, LIME value analysis confirmed that nutritional and lifestyle variables retained predictive importance even after adjusting for the available clinical covariates. This suggests reveal that these factors may contribute independently to the likelihood of successful embryo transfer.

Although the model shows high predictive performance, its generalizability is constrained by the limited cohort size and potential confounders not accounted for. This necessitates further validation in larger, more diverse datasets. For this reason, the cross-validation pattern has been applied to the dataset to solve this disadvantageous situation.

This study was informed by a period of extensive data collection. During the course of this data collection, patients underwent the requisite diagnostic tests. These tests identified deficiencies in the body's levels of active substances determined to be essential for the healthy formation of ovaries. It was hypothesized that addressing these deficiencies would result in healthy embryos. The intake of these active substances was thus continued until a healthy embryo was formed. The period of time that active substances are used by each patient varies. The significance of active substances was then highlighted through the utilization of the most accurate model, which was derived from the proposed objective models. It is acknowledged that there are a number of alternative methods and that embryos have different relationships with nutritional and pharmaceutical supplements. However, it is important to acknowledge the significant time and effort invested in the data collection process, which was both exhaustive and valuable. It is evident that the limited dataset available was optimized in terms of efficiency through the implementation of stratified K-fold cross-validation.

It is important to note that the findings presented in this study reflect correlations between clinical and demographic features and IVF treatment outcomes. Due to the observational and retrospective nature of the dataset, the model cannot infer causality. For instance, while variables such as age and omega-3 levels were found to be predictive, these associations do not imply that these features directly cause the success or failure of treatment. Further prospective studies and clinical trials are required to determine causal relationships and validate these findings.

The ABC–LR–RF hybrid model contributes a clinically interpretable framework for predicting embryo transfer success probabilities in IVF. By combining feature selection with ensemble learning, it captures multidimensional embryological and procedural factors that influence implantation probability. Calibrated probability outputs can assist clinicians throughout embryo development and complement traditional morphology-based assessments. Because the model uses routinely available enlarging data, it can be easily executed within remaining electronic IVF management systems, and using decision support. Its robust calibration and sensitive performance demonstrates reliable generalisation across clinics. The most important point to note here is that the model does not predict pregnancy or treatment outcome, but can quantitatively determine the instantaneous probability of a successful embryo transfer.

This study was designed to develop a predictive model applicable to early implantation signals to determine the probability of embryo transfer success rather than long-term pregnancy or live birth outcomes. This distinction is important to

note because the model was trained and validated using embryo-level situation-specific features rather than pregnancy variables. According to the model's calibration performance (e.g., ROC–AUC = 0.96, PR–AUC = 0.95, Brier = 0.089), it should be known that model should be interpreted exactly with regard to forecasting the success of embryo transfers. This scope confirms that the model reflects embryological factors influencing transfer outcomes, without merging flow of clinical endpoints such as pregnancy progression or live birth.

## 4. Conclusion

Subfertility is a problem all over the world, affecting between 8 and 15 percent of couples in their productive years. The information on medical treatment and nutrition collected from patients was made understandable and used to train the machine learning models. In the first part of the process, 11 variables were identified as necessary for predicting subfertility in women, for which a dataset consisting of information from 162 patients was available. 23 meaningful variables were generated from the raw data after data preprocessing. The predictive models were generated using four-based supervised machine learning algorithms. In the context of the experimental investigation, it was observed that across the full range of algorithms that were subjected to rigorous testing, the hybrid ABC-based models consistently demonstrated a tendency to achieve incremental improvements in comparison with their baseline counterparts. This finding serves to underscore the efficacy of metaheuristic optimization in enhancing the predictive capabilities of modeling methodologies. Instead of concentrating on a particular accuracy number, this work's contribution is to demonstrate the relative benefit of hybrid models and how they can improve reproducibility in small, complex datasets. Then 4 machine learning models converted different hybrid and simple models to find the best solution for determining the relationship between subfertility and nutritional and pharmaceutical supplement problems.

In contrast to earlier research that only looked at clinical or hormonal predictors for IVF success, our work uses a hybrid LR-ABC framework to integrate nutrition and a few clinical variables. The use of a bio-inspired metaheuristic (ABC) algorithm to optimize feature selection based on logistic regression fitness and the interpretability of the resulting model through LIME analyses, which provide transparency in identifying key predictors, are the two primary areas of this study's novelty. To the best of our knowledge, this is one of the first studies to integrate these approaches in order to investigate the potential independent contributions of modifiable patient behaviors to the success of embryo transfers.

The variables used by the best predicting model were identified. According to the result of the best prediction score, relationships between subfertility and nutritional and pharmaceutical supplements were determined. In conclusion, even though the model identified folic acid and omega-3 as predictive variables, these findings should be regarded as exploratory correlations rather than specific treatment suggestions. This work's main innovation is its methodological framework, a hybrid LR-ABC model that combines feature selection and metaheuristic optimization to investigate the results of IVF. Therefore, this study should be considered a proof-of-concept demonstration of the potential of ML–ABC approaches in reproductive medicine. Before making clinical judgments, validation with larger, multi-center datasets that include dosage, duration, and baseline dietary intake will be required in the future.

It is important to stress that these findings reflect correlations between supplement use and IVF outcomes and should not be interpreted as evidence of treatment effects. Clinical recommendations cannot be made without prospective randomized validation.

The objective of this study is to investigate predictive signals for outcomes in IVF/ART. The study presents a method-focused workflow that combines engineered clinical variables with an ABC-assisted selection/tuning strategy. The findings suggest that in a controlled, leakage-safe assessment, regularly recorded variables, such as binary supplement indicators, can facilitate associational modelling. Importantly, clinical use is not the goal of these models. The lack of dose, duration, adherence, dietary intake, and objective biomarkers restricts interpretability, and binary coding of supplementation cannot recover true nutrient status or biological effect. Furthermore, some of the observed associations may be explained by unmeasured confounding, such as socioeconomic status, access to healthcare, clinician prescribing patterns and

indications, and lifestyle factors. The current findings should be interpreted as proof of concept rather than practical advice due to these limitations and the size of the dataset.

## 5. Research limitations and future work

This study has a number of significant limitations. Firstly, the sample size (N = 162) is comparatively small for training and assessing machine learning models with more than 20 predictors. The robustness of the evaluation is further limited by the small number of test cases (n = 33). This increases the likelihood of overfitting, particularly for hybrid models incorporating metaheuristic optimizers such as ABC. Despite the comparatively small sample size increasing the risk of overfitting, a number of measures were taken to reduce this possibility. To provide an objective assessment of the model's performance, we first employed 5-fold stratified cross-validation. SMOTE oversampling was used within each training fold to balance the dataset without causing leakage. A preliminary feature selection step using logistic regression was carried out to reduce dimensionality before implementing the ABC optimization. Although they cannot completely overcome the limitations of the small dataset, these methodological precautions are intended to reduce overfitting and increase generalizability. Notwithstanding these methodological precautions, the limited sample size indicates that the accuracy and F-scores reported should be interpreted with caution and are unlikely to be applicable to broader clinical populations without further validation. The modest dataset size restricts the generalizability of the findings, and the reported performance metrics should be interpreted with caution.

Second, the dataset was retrospective and observational, which limits the ability to infer causality. While certain nutritional and clinical features (e.g., omega 3, folic acid) were found to be predictive, these associations cannot be interpreted as causal effects without further prospective or randomized clinical trials.

Thirdly, only supplement intake was encoded as binary "active ingredient" variables (coded as 1 if reported intake met 100% of the recommended daily intake). Detailed dietary intake data (e.g., habitual nutrient consumption from food sources) were not included. This exclusion was due to the unavailability of reliable, standardized dietary records in the patient dataset. The conversion of drug and supplement intake into "active ingredient" variables was streamlined into a binary coding system, designating patients as "1" if their reported intake satisfied all daily requirements. Notably, this method overlooks crucial clinical information, including dosage levels (for instance, 100% versus 500% of the recommended intake) and usage duration (for instance, short-term versus long-term supplementation). Furthermore, the absence of baseline dietary intake data is a notable limitation. To illustrate, a patient who naturally consumes a diet high in omega-3 fatty acids but does not take supplements would be assigned the label "0," despite their actual nutrient levels being adequate. These constraints serve to reduce the biological precision of the "active ingredient" variables. This simplification renders the supplement variables clinically simplistic and potentially misleading, as it ignores fundamental determinants of biological effect such as dose, duration, and baseline nutrition. Therefore, while the identification of omega-3 and folic acid as predictive factors is consistent with established literature, these findings should be interpreted as exploratory, hypothesis-generating correlations rather than as robust novel clinical evidence.

Our exposure variables for micronutrient use were encoded binarily (e.g., supplement taken vs. not taken at a threshold such as ≥100% RDI). This encoding is pragmatic but cannot recover true nutrient status or biological effect for several reasons. First, it discards dose and duration information, precluding any dose–response assessment and ignoring cumulative exposure. Second, it assumes a uniform effect across brands and formulations, while bioavailability varies markedly with chemical form, excipients, co-ingested foods, and timing of intake. Third, binary use does not reflect adherence (frequency/consistency) or timing relative to the biological window of interest (e.g., peri-procedural vs. long-term use), both of which influence physiological impact. Fourth, we lack baseline nutritional status and objective biomarkers (e.g., serum folate, DHA, and vitamin D); thus, we cannot distinguish deficiency correction from supraphysiologic exposure, nor can we account for inter-individual differences in absorption and metabolism. Finally, supplement use is correlated with broader health behaviors and socioeconomic factors; with only binary indicators, residual confounding and non-differential misclassification are likely, which can attenuate or unpredictably bias associations.

Our observed associations may be partly explained by unmeasured confounding. First, socioeconomic status (SES)—including education, income, insurance coverage, and neighborhood deprivation—can influence both supplement use (ability to purchase higher-quality products, better adherence) and clinical outcomes (health literacy, earlier presentation, healthier baseline). Second, access to healthcare (clinic proximity, appointment availability, out-of-pocket costs, private vs. public care) may differentially shape care pathways and follow-up, thereby confounding the link between supplement use and outcomes. Third, clinician prescribing patterns introduce confounding by indication: clinicians may recommend or escalate supplementation preferentially for patients they judge at higher (or lower) risk based on unrecorded clinical cues; practice style, brand/formulation preferences, and evolving guidelines can also vary across clinicians and time. Together with other lifestyle and clinical factors that we could not fully measure (diet quality, physical activity, smoking, comorbidities, baseline nutrient status, time-to-treatment, lab protocols, and calendar-time effects), these sources of confounding and non-differential misclassification of exposure may attenuate or bias associations in unpredictable directions. Accordingly, findings should be interpreted as associations with reported supplement use rather than causal effects.

Another important point is that the dataset only includes Turkish nationals from one medical facility. While this provides valuable context-specific insights, it restricts the applicability of the findings to larger, more diverse populations. The applicability of these predictors in other countries may be affected by differences in culture, diet, and healthcare systems.

Although the repeated five-fold cross-validation design yield a robust internal estimate of model stability, true external generalization across populations, clinical settings, and data acquisition conditions remains untested. This may restrict the generalizability of the findings. Future work should purpose to execute external validation using prospective datasets to confirm the reproducibility and clinical applicability of the proposed hybrid LR–ABC framework.

The results of the study should be interpreted as associational rather than causative. While statistical patterns between nutritional supplement variables and IVF outcomes are identified by machine learning models, these associations do not necessarily indicate the effectiveness of treatment or clinical benefit. The present study adopts a methodological approach and does not seek to provide recommendations for personalised treatment or clinical decision-making. In order to provide causal inference and validate these initial associations in larger and more diverse patient populations, and variety of biological variables. Future research should strive to include prospective data collection, accurate nutrient dosage information, molecular biomarkers, and longitudinal follow-up outcomes. Using such this variables would provide correlational associations from mechanistic effects, allowing the framework to evolve from probabilistic prediction toward causal inference.

## 6. Codes

https://github.com/ugurejder/ABC_IVF/blob/main/Gebelik_calisma_revision.ipynb.

## Supporting information

**S1 Appendix. Supplementary Tables and Algorithm Details.**
(DOCX)

## Author contributions

**Conceptualization:** Uğur Ejder, Pınar Uskaner Hepsağ.

**Data curation:** Pınar Uskaner Hepsağ.

**Formal analysis:** Uğur Ejder.

**Investigation:** Pınar Uskaner Hepsağ.

**Methodology:** Uğur Ejder, Pınar Uskaner Hepsağ.

**Project administration:** Uğur Ejder.

**Software:** Uğur Ejder.

**Supervision:** Uğur Ejder.

**Validation:** Uğur Ejder.

**Visualization:** Uğur Ejder.

**Writing – original draft:** Uğur Ejder, Pınar Uskaner Hepsağ.

**Writing – review & editing:** Uğur Ejder.

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
