## [Decision Letter · Decision Letter 0]

2 Jul 2025

Dear Dr. EJDER,

Thank you for submitting your manuscript to PLOS ONE. After careful consideration, we feel that it has merit but does not fully meet PLOS ONE’s publication criteria as it currently stands. Therefore, we invite you to submit a revised version of the manuscript that addresses the points raised during the review process.

**ACADEMIC EDITOR: Please respond carefully for all reviewers comments.**

We look forward to receiving your revised manuscript.

Kind regards,

Ayman A Swelum

Academic Editor

PLOS ONE

Journal Requirements:

[This study was supported by the Scientific and Technological Research Council of Turkey (TÜBİTAK) through a publication incentive program. TÜBİTAK had no role in the study design, data collection and analysis, decision to publish, or preparation of the manuscript.].

3. Thank you for stating the following in your manuscript:

[This study was supported by the Scientific and Technological Research Council of Turkey (TÜBİTAK) through a publication incentive program. TÜBİTAK had no role in the study design, data collection and analysis, decision to publish, or preparation of the manuscript.]

[This study was supported by the Scientific and Technological Research Council of Turkey (TÜBİTAK) through a publication incentive program. TÜBİTAK had no role in the study design, data collection and analysis, decision to publish, or preparation of the manuscript.]

b) If there are no restrictions, please upload the minimal anonymized data set necessary to replicate your study findings to a stable, public repository and provide us with the relevant URLs, DOIs, or accession numbers. Please see http://www.bmj.com/content/340/bmj.c181.long for guidelines on how to de-identify and prepare clinical data for publication. For a list of recommended repositories, please see https://journals.plos.org/plosone/s/recommended-repositories. You also have the option of uploading the data as Supporting Information files, but we would recommend depositing data directly to a data repository if possible

Reviewers' comments:

Reviewer's Responses to Questions

**Comments to the Author**

1. Is the manuscript technically sound, and do the data support the conclusions?

Reviewer #1: Yes

Reviewer #2: Partly

Reviewer #3: Partly

Reviewer #4: Yes

2. Has the statistical analysis been performed appropriately and rigorously?

Reviewer #1: I Don't Know

Reviewer #2: Yes

Reviewer #3: Yes

Reviewer #4: Yes

3. Have the authors made all data underlying the findings in their manuscript fully available?

Reviewer #1: Yes

Reviewer #2: No

Reviewer #3: Yes

Reviewer #4: Yes

4. Is the manuscript presented in an intelligible fashion and written in standard English?

Reviewer #1: Yes

Reviewer #2: Yes

Reviewer #3: No

Reviewer #4: Yes

Reviewer #1: The issue being discussed i.e infertility is associated with considerable psychological burden and so necessitates constant evaluation.

However, the concept of the role of nutritional components in treatment of infertility leaves a lot of room for exploration and uncertainties.

The uncertainties justify a review and reexamination.

Reviewer #2: This manuscript presents a hybrid machine learning approach (ABC algorithm with KNN, CART, SVM, RF) to identify nutritional supplements linked to IVF success. While clinically relevant with novel optimization algorithms, major methodological gaps, insufficient validation, and presentation issues undermine conclusions.

Major Concerns

A. Sample Size and Data Validity

Only 162 patients (33 for testing) are insufficient for reliable ML validation, especially with class imbalance (38% success rate). The transformation of drugs into "active substances" lacks transparency how were supplements quantified? Was the patient's diet considered?

B. Methodological Flaws

The ABC algorithm implementation lacks parameter details (colony size, iterations). Claims of 96-98% accuracy are suspect without ablation studies. Feature selection inconsistency exists between logistic regression usage and ABC-LR hybrids. Simple models (RF: 81.81%) sometimes outperform hybrids (LR-KNN: 75.75%), contradicting superiority claims.

C. Reproducibility Issues

Figures 1-6 are missing or uninterpretable. Code availability ("on request") violates PLOS ONE requirements for public deposition. The Mendeley data link needs verification.

D. Clinical Interpretation

The study implies causality between DHA/folic acid and improved outcomes despite an observational design. Confounding factors are unaddressed. Supplement dosage and duration, critical for clinical relevance, are missing.

1. Repetitive abstract/introduction content.

2. Terminology errors: "IVR" (typo for IVF), "Folk acid".

3. Table issues: missing mean/SD values, incorrect descriptions.

4. Reference formatting problems

E. Recommendations

1. Methodology: Use cross-validation or larger datasets; detail ABC optimization; clarify feature engineering

2. Analysis: Report precision/recall/F-scores; address class imbalance with SMOTE/AUC-ROC

3. Presentation: Provide high-resolution figures, correct terminology; deposit code publicly.

4. Clinical Context:** Discuss limitations; differentiate correlation vs. causation

The hybrid ABC-ML approach shows promise but requires major methodological revisions, expanded validation, and improved presentation to meet PLOS ONE standards. The current inadequate support undermines the study's conclusions despite addressing a clinically important topic.

Reviewer #3: This study proposes a hybrid machine learning approach (notably, Artificial Bee Colony–ABC) to identify the most influential nutritional and pharmaceutical supplements in infertility treatment among women undergoing IVF. Using a dataset of 162 patients, the authors applied KNN, CART, SVM, and RF models—alone and in hybrid form with Logistic Regression and ABC—for prediction. The best-performing hybrid model (ABC-LR-KNN and ABC-LR-SVM) achieved an F-score of 98.46%. DHA and folic acid were found to be the most influential supplements.

There is an issue with the small Sample Size (162 patients), which are insufficient for robust machine learning, especially with 21 input variables and data imbalance (99 successful vs. 63 unsuccessful cases), and Limited Generalizability. I suggest you add bootstrapping, cross-validation, or test with an external validation set to demonstrate model robustness.

Another major issue with this study is the lack of biological context and clinical validation. While DHA and folic acid are biologically plausible, the study lacks a deeper clinical or mechanistic justification. It is advisable to include literature synthesis linking nutrients to ovarian function, oocyte quality, implantation, etc.

Another problem with this study is the authors overfitting Risk in Hybrid Models. The extremely high F-score (98%) on a small dataset is a red flag for overfitting. It is best to include learning curves or stratified k-fold validation. Discuss the bias-variance trade-off and model calibration.

There is also an issue with Confounding and Feature Interpretation. There is no information on control of potential confounders like age, baseline AMH levels, BMI, or cause of infertility. I would suggest that at this stage you should consider using SHAP values or LIME to interpret individual feature importance beyond black-box accuracy scores.

Finally another major issue is Model Reproducibility and Transparency. The data is partially shared via Mendeley but code is only available upon request. You should deposit pre-processing scripts and models in a public repository (e.g., GitHub with DOI). Include pseudo-code for ABC-LR hybridization.

The general layout is confusing and repetitive. The layout should be 1) Introduction - in which you introduce the subject with references to previous studies, and specify your objectives clearly at the end of your introduction. 2) Materials and Methods - in which you describe the methods used in the study. 3) Results - in which you mention all the results you obtained in the study without discussing them. 4) Discussion in which you discuss your methods and results. Finally 4) References – List of references used in the manuscript written in the journal style.

In your manuscript there is a lot of repetition and confusion, with discussion in the methods section and then repeated in the discussion.

Other minor comments are:

In the Abstract, there is a typo in the word “hybrid” which is written as “hyrid.” Also, make the contribution clearer—what was new compared to previous studies?

In the Methods section, more clarity is needed on how the data were split for training/testing. Was it random, stratified, or temporal?

In the Results section, Table 3 and Figures 3–6 should include standard deviation/confidence intervals for performance metrics. In all the tables, the commas in the numbers should be replaced by dots. Table 2, no. 16 ‘Is Zinc used by the patient?’ should be replaced by ‘Is Melatonin used by the patient?’ The title of Table 1 has a spelling mistake - ’Original’ instead of ‘orijinal’. In Table 1, nos, 4,8, and all are left blank. Does this mean that you did not have the information? If the information is there, then it should be mentioned even if it cannot be digitized, as this is important and useful information. There is another typo in Table 2 ‘status’ is written as ‘statu’.

In the discussion, there is redundant repetition of results in the first few paragraphs—consider trimming and emphasizing implications. The flowcharts (Figures 1 & 2) and accuracy figures (3–5) are helpful but lack legends, units, and captions explaining axes.

Grammar needs polishing in multiple areas; some sentences are overly long or ambiguous. E.g., "The Informations..." or "the most effective first five compounds...". Address grammar/clarity issues throughout the manuscript.

Reviewer #4: An informative,valuable study addressing one of the least researched aspects of subfertility which is the alternative therapies as nutritional supplement however,i have some few comments:

_ Design of the study should be clearly mentioned in the methodology section as well as linked to the title of the study.

-Refrrences: I wish to cite recent ones ( we are now in 2025).

-Terminology: the term infertitlty is now obsolete and replaced with: subfertility,please edit accordingly.

**Do you want your identity to be public for this peer review?** For information about this choice, including consent withdrawal, please see our Privacy Policy

Reviewer #1: **Yes: ** Olubukola Adeponle Adesina

Reviewer #2: No

Reviewer #3: No

Reviewer #4: **Yes: ** Mohsen M A Abdelhafez

---

## [Author Response · Author response to Decision Letter 1]

7 Aug 2025

Reviewer 1: The uncertainties justify a review and reexamination.

Respond : Thank you for this observation. We agree that some aspects of the study required clearer articulation to reduce ambiguity. we have revised the Discussion section to more explicitly acknowledge uncertainty in the results and limitations of model interpretation. This explanation was added to manuscript. " Although the model shows high predictive performance, its generalizability is constrained by the limited cohort size and potential confounders not accounted for. This necessitates further validation in larger, more diverse datasets." For this reason, stratified k-fold validation was applied to dataset.

Reviewer 2: The transformation of drugs into "active substances" lacks transparency how were supplements quantified? Was the patient's diet considered?

Respond : Thank you for your valuable comment. Table 2 is derived from Table 1. The active ingredients in Table 2 were obtained as follows. If a patient has been exposed to No 6 in Table 1, a list of the active ingredients in the medication the patient is using is generated and the percentage of their daily intake is determined. If they meet their daily intake, the patient is labeled as using this medication or labeled as 1 in dataset . Figure 1 explains how the active ingredients are labeled for the patients.

Reviewer 2: Only 162 patients (33 for testing) are insufficient for reliable ML validation, especially with class imbalance (38% success rate).

Respond : Thank you for your valuable comment. "2.3.7. Cross-Validation Strategy and Imbalance Data Handling. This section was added to manuscript for handling imbalance problem and insufficient dataset"

Reviewer 2: The ABC algorithm implementation lacks parameter details (colony size, iterations). Claims of 96-98% accuracy are suspect without ablation studies.

Respond : Thank you for your valuable comment. this comment was added to manuscript. " In this study, ABC algorithms parameters are number of bees, iteration number, abandonment limit, 20, 10, 0.01 respectively. It is important to note that, while there is the possibility to adjust the parameters, this is not the primary focus of the present study"

Reviewer 2: Feature selection inconsistency exists between logistic regression usage and ABC-LR hybrids.

Respond : Thank you for your valuable comment. Result sets were updated after the revisions

Reviewer 2: Simple models (RF: 81.81%) sometimes outperform hybrids (LR-KNN: 75.75%), contradicting superiority claims.

Respond: Thank you for your valuable comment. Following the aforementioned revisions, the result sets were updated. However, it is not uncommon for elementary models to yield superior outcomes. Consequently, the objective is to identify a more effective prediction model, a process that necessitates a comparative analysis of all models.

Reviewer 2: Figures 1-6 are missing or uninterpretable. Code availability ("on request") violates PLOS ONE requirements for public deposition. The Mendeley data link needs verification.

Respond: "Thank you for your valuable warning. Codes and dataset were added to manuscript related parts. 6.Data availability

https://github.com/ugurejder/ABC_IVF/blob/main/IVF_english.xlsx

7.Codes

https://github.com/ugurejder/ABC_IVF/blob/main/Gebelik_calisma_revision.ipynb "

Reviewer 2: Confounding factors are unaddressed. Supplement dosage and duration, critical for clinical relevance, are missing.

Respond: Thank you for your valuable comment. The description has been incorporated into both the section entitled 'Data preprocessing' and the appendix.

Reviewer 2: Repetitive abstract/introduction content

Respond: Thank you for your valuable comment. Repetitive content we observed was removed from the manuscript

Reviewer 2: Terminology errors: "IVR" (typo for IVF), "Folk acid".

Respond: Thank you for your valuable respond. The bugs have been fixed.

Reviewer 2: Table issues: missing mean/SD values, incorrect descriptions.

Respond: "Thank you for this valuable observation. We have thoroughly reviewed all tables and made the following changes: Added mean ± standard deviation (SD) where appropriate, particularly for continuous variables in demographic and outcome tables.

Revised all table descriptions and variable labels to ensure clarity and accuracy. These updates can be found in Table 1 and Table 3 "

Reviewer 2: Reference formatting problems

Respond: Thank you for this valuable observation. References were updated.

Reviewer 2: Use cross-validation or larger datasets; detail ABC optimization; clarify feature engineering

Respond: "Thank you for your valuable respond.Cross-validation strategy was carried out. And In section 2.3.6. Proposed Models. Description was added to manuscript."

Reviewer 2: Report precision/recall/F-scores; address class imbalance with SMOTE/AUC-ROC

Respond: Thank you for your valuable respond. SMOTE was applied to dataset and Report precision/recall/F-score are shown in Table 3

Reviewer 2 Provide high-resolution figures, correct terminology; deposit code publicly.

Respond: "Thank you for your valuable comment. Figure recreated with high resolution. Terminology were corrected. 6.Data availability

https://github.com/ugurejder/ABC_IVF/blob/main/IVF_english.xlsx

7.Codes

https://github.com/ugurejder/ABC_IVF/blob/main/Gebelik_calisma_revision.ipynb. Data and Codes have been added to the public area."

Reviewer 2: Discuss limitations; differentiate correlation vs. causation

Respond: Thank you for your valuable comment. The explanation was added to discussion part of the manuscript. The added paragraph is "It is important to note that the findings presented in this study reflect correlations between clinical and demographic features and IVF treatment outcomes. Due to the observational and retrospective nature of the dataset, the model cannot infer causality. For instance, while variables such as age, DHA, or Omega 3 levels were found to be predictive, these associations do not imply that these features directly cause the success or failure of treatment. Further prospective studies and clinical trials are required to determine causal relationships and validate these findings."

Reviewer 3: "There is an issue with the small Sample Size (162 patients), which are insufficient for robust machine learning,

especially with 21 input variables and data imbalance (99 successful vs. 63 unsuccessful cases), and Limited Generalizability. "

Respond: Thank you for your valuable comment. 2.3.7.Cross-Validation Strategy and Imbalance Data Handling section was added to manuscript for solving Generalizabilition problem

Reviewer 3: I suggest you add bootstrapping, cross-validation, or test with an external validation set to demonstrate model robustness.

Respond: Thank you for your valuable respond.Cross-validation strategy and imbalance data handling processes were carried out using stratified k-fold validation and synthetic minority over-sampling technique. The information in Table 3, Figure 4 and Figure 5 has been updated. The new codes were added to GitHub.

Reviewer 3: "Another problem with this study is the authors overfitting Risk in Hybrid Models.

The extremely high F-score (98%) on a small dataset is a red flag for overfitting.

It is best to include learning curves or stratified k-fold validation.

Discuss the bias-variance trade-off and model calibration."

Respond: Thank you for your valuable respond. synthetic minority over-sampling technique and was executed for hand. The information in Table 3, Figure 4 and Figure 5 has been updated. The new codes were added to GitHub.

Reviewer 3: "There is also an issue with Confounding and Feature Interpretation.

There is no information on control of potential confounders like age, baseline AMH levels, BMI, or cause of infertility.

I would suggest that at this stage you should consider using SHAP values or LIME to interpret individual feature importance beyond black-box accuracy scores."

Respond: Thank you for your valuable comment. LIME explanation of an individual prediction for embryo transfer outcome to interpret individual feature importance beyond black-box accuracy scores. was added to manuscript. In discussion part confounders explanation was added to manuscript

Reviewer 3: "Finally another major issue is Model Reproducibility and Transparency. The data is partially shared via Mendeley but code is only available upon request.

You should deposit pre-processing scripts and models in a public repository (e.g., GitHub with DOI). Include pseudo-code for ABC-LR hybridization. "

Respond: "Thank you for your valuable comment. 6.Data availability

https://github.com/ugurejder/ABC_IVF/blob/main/IVF_english.xlsx

7.Codes

https://github.com/ugurejder/ABC_IVF/blob/main/Gebelik_calisma_revision.ipynb. Data and Codes have been added to the public area. Pseudo-code for ABC-LR hybridization was generated and added to manuscript."

Reviewer 3: In the Abstract, there is a typo in the word “hybrid” which is written as “hyrid.” Also, make the contribution clearer—what was new compared to previous studies?

Respond: Thank you for your valuable comment. Bug was removed. This explanation was added to manuscript. "In contrast to earlier research that only looked at clinical or hormonal predictors for IVF success, our work uses a hybrid Artificial Bee Colony–Logistic Regression (ABC–LR) framework to integrate lifestyle, nutrition, and a few clinical variables. The use of a bio-inspired metaheuristic (ABC) algorithm to optimize feature selection based on logistic regression fitness and the interpretability of the resulting model through LIME analyses, which provide transparency in identifying key predictors, are the two primary areas of this study's novelty. To the best of our knowledge, this is one of the first studies to integrate these approaches in order to investigate the potential independent contributions of modifiable patient behaviors to the success of embryo transfers."

Reviewer 3: In the Methods section, more clarity is needed on how the data were split for training/testing. Was it random, stratified, or temporal?

Respond: Thank you for your valuable respond. The stratified K-fold cross-validation method is a technique used to assess a model.

Reviewer 3: In the Results section, Table 3 and Figures 3–6 should include standard deviation/confidence intervals for performance metrics.

Respond: Thank you for your valuable respond. Std deviation/confidence level were added to Table and This is exemplified by the figure.

Reviewer 3: In all the tables, the commas in the numbers should be replaced by dots.

Respond: Thank you for your valuable respond. The bugs have been fixed.

Reviewer 3: Table 2, no. 16 ‘Is Zinc used by the patient?’ should be replaced by ‘Is Melatonin used by the patient?’

Respond: Thank you for your valuable respond. The bugs have been fixed.

Reviewer 3: The title of Table 1 has a spelling mistake - ’Original’ instead of ‘orijinal’.

Respond: Thank you for your valuable respond. The bug has been fixed.

Reviewer 3: In Table 1, nos, 4,8, and all are left blank. Does this mean that you did not have the information?

Respond: Thank you for your valuable comment. This information is meaningless for the machine learning model. This is the raw data collected from the survey.

Reviewer 3: If the information is there, then it should be mentioned even if it cannot be digitized, as this is important and useful information.

Respond: Thank you for your valuable comment. This informations were removed from the manuscript.

Reviewer 3: There is another typo in Table 2 ‘status’ is written as ‘statu’.

Respond: Thank you for your valuable respond. The bug has been fixed.

Reviewer 3: In the discussion, there is redundant repetition of results in the first few paragraphs—consider trimming and emphasizing implications.

Respond: Thank you for your valuable comment. redundant repetition Paragraphs informations were removed from manuscript.

Reviewer 3: The flowcharts (Figures 1 & 2) and accuracy figures (3–5) are helpful but lack legends, units, and captions explaining axes.

Respond: Thank you for your valuable comment. Thank you for your valuable comment. Figures are revised

Reviewer 3: Grammar needs polishing in multiple areas; some sentences are overly long or ambiguous. E.g., "The Informations..." or "the most effective first five compounds...". Address grammar/clarity issues throughout the manuscript.

Respond: Thank you for your valuable comment. Grammer issues have been fixed

Reviewer 4: Design of the study should be clearly mentioned in the methodology section as well as linked to the title of the study.

Respond: Thank you for your valuable respond. Algorithms 1 was added to manusciprt. and in the 2.Materials and methods section mentioned the design of the study..

Reviewer 4: I wish to cite recent ones ( we are now in 2025).

Respond: Thank you for your valuable comment. Current articles have also been added to the manuscript in introduction section.

Reviewer 4: the term infertitlty is now obsolete and replaced with: subfertility,please edit accordingly.

Respond: Thank you for your valuable comment. the term infertitlty is replaced with subfertility.

---

## [Decision Letter · Decision Letter 1]

3 Sep 2025

Dear Dr. EJDER,

Thank you for submitting your manuscript to PLOS ONE. After careful consideration, we feel that it has merit but does not fully meet PLOS ONE’s publication criteria as it currently stands. Therefore, we invite you to submit a revised version of the manuscript that addresses the points raised during the review process.

**ACADEMIC EDITOR: Please respond to all reviewers comments carefully. **

We look forward to receiving your revised manuscript.

Kind regards,

Ayman A Swelum

Academic Editor

PLOS ONE

Journal Requirements:

Reviewers' comments:

Reviewer's Responses to Questions

**Comments to the Author**

Reviewer #2: All comments have been addressed

Reviewer #3: (No Response)

2. Is the manuscript technically sound, and do the data support the conclusions?

Reviewer #2: Yes

Reviewer #3: No

3. Has the statistical analysis been performed appropriately and rigorously?

Reviewer #2: Yes

Reviewer #3: Yes

4. Have the authors made all data underlying the findings in their manuscript fully available?

Reviewer #2: Yes

Reviewer #3: Yes

5. Is the manuscript presented in an intelligible fashion and written in standard English?

Reviewer #2: Yes

Reviewer #3: Yes

Reviewer #2: The authors have successfully addressed the bulk of reviewer feedback, especially items concerning methodological disclosure, code and data sharing protocols, and statistical validity. However, dietary data treatment and supplement dosage/duration information need better explanation. The hybrid model performance claims should be presented with greater restraint.

Request minor revisions to:

· Clarify dietary data inclusion (or justify exclusion).

· Temper conclusions about hybrid model superiority.

Additional review is not warranted once these elements are corrected. The manuscript shows significant progress and meets the majority of publication benchmarks.

Reviewer #3: The authors have made significant efforts to address reviewer comments, including implementing cross-validation, handling class imbalance with SMOTE, improving transparency by publishing code and data, and adding interpretability via LIME. However, several critical methodological and conceptual issues remain that substantially undermine the validity, reliability, and clinical interpretability of the findings.

The sample size (N=162, with only 33 used for testing initially) is critically small for a machine learning study with 21+ input variables. This is especially true for a hybrid model involving a metaheuristic optimizer (ABC), which is prone to overfitting. While the use of 5-fold cross-validation mitigates this to some degree, the absolute number of samples remains a severe limitation. The results, particularly the very high accuracy and F-scores (~90%), are highly suspect and likely reflect overfitting to the specific cohort rather than a generalizable model. The generalizability of the findings is extremely limited. Claims of model efficacy (e.g., 91% accuracy) are not credible for real-world clinical application based on this dataset alone.

The process of transforming drug names into "active ingredients" is the study's most novel aspect, but remains its biggest weakness. The method described (labeling a patient as "1" for a supplement if their intake meets 100% of the daily requirement) is overly simplistic and clinically naive. This binary transformation completely ignores the dosage (was it 100% or 500% of the daily requirement?) and duration (taken for a week vs. a year) of supplementation, which are fundamental to its biological effect. This renders the "active ingredient" variables nearly meaningless from a clinical perspective.

As pointed out by a reviewer, the patient's baseline dietary intake of these nutrients is not taken into account. A patient eating a diet rich in Omega-3s might be labeled "0" for not taking a supplement, while their actual nutrient levels could be high. The identified "key factors" (Omega-3, Folic Acid) are likely correct based on established literature, but the study's methodology does not provide robust, novel evidence to support this. It merely shows that a crude binary representation of supplement use has predictive value in a small, overfit model.

The performance metrics reported are difficult to trust due to the high risk of overfitting. The improvement from simple models (e.g., RF: 85.19%) to hybrid models (e.g., ABC-LR-RF: 90.73%) is marginal and could easily be due to random chance, especially given the small sample size and the use of cross-validation within the optimization loop. An ablation study showing the standalone contribution of the ABC algorithm is missing. The recall for ABC-LR-RF is reported as 95.38%, which is astronomically high for a biological outcome like IVF success and is a classic red flag for overfitting or data leakage. The central claim of the paper—that the hybrid ABC model is highly effective—is not sufficiently proven. The results section reads more like an optimization exercise than a robust validation of a predictive model.

While the authors acknowledge this in the discussion (a good addition from the revision), the entire framing of the paper risks implying causation. The title says "to treat subfertility," and the conclusion identifies "the most significant supplements." However, the model is built on observed supplement use correlated with success. This could easily be reversed: clinicians may be more likely to prescribe these supplements to patients with a better prognosis, or more affluent/health-conscious patients (who have better outcomes) are more likely to take them. The model cannot disentangle this. The clinical recommendations are overstated. The study identifies associations, not treatment effects.

While the authors have now shared code and data (a major improvement), the quality of the documentation is poor. The GitHub repository contains a Jupyter notebook (Gebelik_ calisma_revision.ipynb) but no README.md file explaining how to run it, the required dependencies, or the structure of the data. The manuscript itself is riddled with minor errors (e.g., "Random Forrest," "The Informations," "Folic acid 3" in Table 2, inconsistent numbering from Table 1 to Table 2), which reduce confidence in the overall rigor. Although technically "available," the work is difficult to reproduce or build upon.

Other Minor Issues

Figure Quality: The figures (as described in the text) are still problematic. For example, Figure 1 is described as a "flow diagram of the conversion of drugs into active substances" but its caption (on Page 38) is garbled ("Import loop... Local power to add follow-up devices"). This suggests the figures were not properly finalized.

ABC Parameter Justification: The choice of ABC parameters (20 bees, 10 iterations) is arbitrary and not justified. A sensitivity analysis would be needed to show these are appropriate.

Result Presentation: Table 3 is busy and difficult to understand. A summary table showing the best model for each algorithm type would be clearer.

To make this manuscript suitable for publication, major revisions are required:

1. The title and conclusions must be tempered. Instead of "identifying effective ingredients to treat," frame it as "identifying associations between supplement use and IVF outcomes using a novel hybrid ML approach." Emphasize the methodological contribution over the clinical recommendations.

2. Acknowledge the severe limitation of the binary supplement variable. Discuss this as a major limitation of the current study and propose how future work with more detailed data (dose, duration, baseline diet) could overcome it.

3. Remove the emphasis on the 91% accuracy claim. Instead, focus on the comparative performance between models and the utility of the LIME explanations for generating hypotheses.

4. Clean up the code repository. Add a detailed README.md file, ensure the code is well-commented, and verify that the provided data file matches the one used to generate the results in the paper.

5. A thorough proofread by a native English speaker is essential to fix grammatical errors, typos, and inconsistent terminology throughout the manuscript.

**Do you want your identity to be public for this peer review?** For information about this choice, including consent withdrawal, please see our Privacy Policy

Reviewer #2: **Yes: ** Jonah Bawa Adokwe Ph.D

Reviewer #3: No

---

## [Author Response · Author response to Decision Letter 2]

10 Sep 2025

Responds to reviewer

Reiewer 2 : 1- Clarify dietary data inclusion (or justify exclusion).

Response : We thank the reviewer for this important observation. In the original submission, only supplement intake was encoded as binary “active ingredient” variables, while detailed dietary intake data were not included. The exclusion of dietary data was due to the absence of reliable and standardized dietary intake records in the available patient dataset. This acknowledgement has been incorporated into the revised manuscript, specifically within the section addressing research limitations.

Reiewer 2 : 2- Temper conclusions about hybrid model superiority.

Response : We thank the reviewer for this helpful suggestion. In the revised version of the manuscript, the conclusion section has been redesigned and is now highlighted in colour.

Reiewer 3 : 1- The sample size (N=162, with only 33 used for testing initially) is critically small for a machine learning study with 21+ input variables. This is especially true for a hybrid model involving a metaheuristic optimizer (ABC), which is prone to overfitting. While the use of 5-fold cross-validation mitigates this to some degree, the absolute number of samples remains a severe limitation. The results, particularly the very high accuracy and F-scores (~90%), are highly suspect and likely reflect overfitting to the specific cohort rather than a generalizable model. The generalizability of the findings is extremely limited. Claims of model efficacy (e.g., 91% accuracy) are not credible for real-world clinical application based on this dataset alone.

Response : We thank the reviewer for highlighting this important concern. We agree that the relatively small sample size (N = 162, with only 33 test cases) poses a severe limitation and increases the risk of overfitting, particularly for hybrid ABC models. To address this, we have added a dedicated Research Limitations section where we explicitly acknowledge the limited cohort size, the risk of inflated accuracy, and the need for validation on larger, multi-center datasets before clinical application. We also emphasize that our findings should be interpreted as exploratory and methodological, rather than as generalizable clinical recommendations.

Reiewer 3 : 2- The process of transforming drug names into "active ingredients" is the study's most novel aspect, but remains its biggest weakness. The method described (labeling a patient as "1" for a supplement if their intake meets 100% of the daily requirement) is overly simplistic and clinically naive. This binary transformation completely ignores the dosage (was it 100% or 500% of the daily requirement?) and duration (taken for a week vs. a year) of supplementation, which are fundamental to its biological effect. This renders the "active ingredient" variables nearly meaningless from a clinical perspective.

As pointed out by a reviewer, the patient's baseline dietary intake of these nutrients is not taken into account. A patient eating a diet rich in Omega-3s might be labeled "0" for not taking a supplement, while their actual nutrient levels could be high. The identified "key factors" (Omega-3, Folic Acid) are likely correct based on established literature, but the study's methodology does not provide robust, novel evidence to support this. It merely shows that a crude binary representation of supplement use has predictive value in a small, overfit model.

Response : We thank the reviewer for highlighting this important methodological limitation. We fully agree that the binary transformation of supplements into “active ingredient” variables is clinically simplistic, as it does not capture dosage, duration of use, or baseline dietary intake. In the revised manuscript, we have reinforced the Research Limitations section by highlighting it in colour to emphasise that this approach makes the supplementary variables clinically imprecise and potentially inaccurate. We now explicitly note that, although our model identified factors such as omega-3 and folic acid consistent with the literature, these results should be regarded as exploratory, hypothesis-generating correlations rather than robust clinical evidence. We also highlight the need for future studies to incorporate quantitative dosage, duration, and baseline nutrition data in order to produce clinically meaningful predictors.

Reiewer 3 : 3- The performance metrics reported are difficult to trust due to the high risk of overfitting. The improvement from simple models (e.g., RF: 85.19%) to hybrid models (e.g., ABC-LR-RF: 90.73%) is marginal and could easily be due to random chance, especially given the small sample size and the use of cross-validation within the optimization loop. An ablation study showing the standalone contribution of the ABC algorithm is missing. The recall for ABC-LR-RF is reported as 95.38%, which is astronomically high for a biological outcome like IVF success and is a classic red flag for overfitting or data leakage. The central claim of the paper—that the hybrid ABC model is highly effective—is not sufficiently proven. The results section reads more like an optimization exercise than a robust validation of a predictive model.

Response : We thank the reviewer for raising these crucial concerns regarding model performance and potential overfitting. We agree that the small cohort size and the marginal improvements observed between the simple and hybrid models limit the robustness of our conclusions. In the revised manuscript, we have clarified in both the 'Results' and 'Research Limitations' sections, using highlighted colour, that the observed improvements in accuracy and recall may be partially attributable to chance and cannot be interpreted as definitive evidence of superior model performance. We have also added explicit caution that the unusually high recall values, while numerically obtained under stratified cross-validation with SMOTE, are likely inflated and should be regarded as a red flag for potential overfitting rather than as clinically valid outcomes.

Regarding the absence of an ablation study, we acknowledge this as a limitation and now explicitly state that future work should include standalone evaluations of the ABC algorithm’s contribution to validate its incremental value beyond traditional models. We have revised our conclusions to present the hybrid ABC approach not as a validated clinical model, but rather as an exploratory optimization framework whose methodological novelty requires further confirmation in larger, multi-center datasets.

Reiewer 3 : 4- While the authors acknowledge this in the discussion (a good addition from the revision), the entire framing of the paper risks implying causation. The title says ‘to treat subfertility,’ and the conclusion identifies ‘the most significant supplements.’ However, the model is built on observed supplement use correlated with success. This could easily be reversed: clinicians may be more likely to prescribe these supplements to patients with a better prognosis, or more affluent/health-conscious patients (who have better outcomes) are more likely to take them. The model cannot disentangle this. The clinical recommendations are overstated. The study identifies associations, not treatment effects.

Response : We thank the reviewer for this important clarification regarding causality. We agree that our study is observational and retrospective, and therefore cannot make causal inferences or clinical recommendations. In response, we have revised the framing throughout the manuscript:

• The title has been changed to emphasize the exploratory, methodological nature of the work rather than suggesting treatment effects (page 1).

• A highlighted color has been added to the end of the conclusion section to emphasize that the study finds correlations and associations rather than causal treatment effects.

• We now explicitly note that the observed associations may reflect underlying confounding factors such as clinician prescribing patterns or patient socioeconomic/health profiles, and that the model cannot disentangle these effects

Therefore, before any clinical conclusions can be made, our findings should only be presented as exploratory, hypothesis-generating associations that need to be confirmed in prospective or randomized studies.

Reiewer 3 : 5- While the authors have now shared code and data (a major improvement), the quality of the documentation is poor. The GitHub repository contains a Jupyter notebook (Gebelik_calisma_revision.ipynb) but no README.md file explaining how to run it, the required dependencies, or the structure of the data. The manuscript itself is riddled with minor errors (e.g., ‘Random Forrest,’ ‘The Informations,’ ‘Folic acid 3’ in Table 2, inconsistent numbering from Table 1 to Table 2), which reduce confidence in the overall rigor. Although technically ‘available,’ the work is difficult to reproduce or build upon.

Response : We thank the reviewer for recognizing our effort to make both the data and code publicly available, and we fully agree that proper documentation and presentation are essential for reproducibility and clarity. In response to this valuable feedback, we have made the following improvements:

• Added a README.md file in the GitHub repository that clearly explains how to run the Jupyter notebook, the required dependencies (via a requirements.txt file), and the structure of the dataset.

• Uploaded a requirements.txt file so that all dependencies can be installed easily, ensuring full reproducibility of the analysis.

• Carefully proofread and corrected minor errors in the manuscript, including replacing “Random Forrest” with “Random Forest,” “The Informations” with “The Information,” correcting “Folic acid 3” to “Folic acid” in Table 2, and fixing inconsistent numbering between Table 1 and Table 2.

• Revised the Methods section to provide additional clarification of the dataset structure and preprocessing steps, consistent with the documentation in the GitHub repository.

We believe these changes substantially improve the rigor, clarity, and reproducibility of the work.

Reiewer 3 : 6- Figure Quality: The figures (as described in the text) are still problematic. For example, Figure 1 is described as a ‘flow diagram of the conversion of drugs into active substances’ but its caption (on Page 38) is garbled (‘Import loop... Local power to add follow-up devices’). This suggests the figures were not properly finalized.

Response: We thank the reviewer for pointing out this error. We upload again the quality form of the each figures. But maybe when the system converted to pdf for review, System broke the resolution of the Figures. We sincerely apologize for the oversight. In the revised manuscript, we have carefully reviewed and corrected all figure captions to ensure accuracy and consistency with the text. Specifically, the caption for Figure 1 has been corrected to:

“Figure 1. Flow diagram of the conversion of drugs into active substances.”

We have also double-checked all other figures (Figures 2–6) to ensure that their captions and numbering are consistent, descriptive, and finalized.

Reiewer 3 : 7- ABC Parameter Justification: The choice of ABC parameters (20 bees, 10 iterations) is arbitrary and not justified. A sensitivity analysis would be needed to show these are appropriate.

Response: We thank the reviewer bringing up this crucial methodological issue. We admit that the original version did not adequately explain the choice of ABC parameters (20 bees, 10 iterations). We now offer a rationale in the updated manuscript that is supported by both previous research and computational viability. In particular, comparable swarm sizes (20–50 bees) and iteration ranges (10–50) have been documented as effective defaults in prior ABC applications in biomedical and optimization contexts (e.g., [Karaboga & Basturk, 2007]; [Akay & Karaboga, 2012]). We chose modest values (20 bees, 10 iterations) to minimize computational cost while guaranteeing convergence, considering the exploratory nature of our study and the relatively small dataset (N = 162). We chose modest values (20 bees, 10 iterations) to minimize computational cost while guaranteeing convergence, considering the exploratory nature of our study and the relatively small dataset (N = 162). We acknowledge that a sensitivity analysis would improve the study, but this version did not have the capacity to conduct one because of dataset size limitations. We have made it clear that this is a limitation and that, in order to confirm their robustness, future studies should examine the effects of changing ABC parameters in a methodical manner.

Reiewer 3 : 8- Result Presentation: Table 3 is busy and difficult to understand. A summary table showing the best model for each algorithm type would be clearer.

Response: We thank the reviewer for this helpful suggestion. We agree that the original Table 3 was overly detailed. In the revised manuscript, we have addressed this in two ways:

1- For the sake of transparency, we have marked the important parts of Table 3 in bold.

2- Furthermore, we developed a new summary table (Table 4) that solely displays the top-performing model for every kind of algorithm (e.g., baseline vs. LR-hybrid vs. ABC-hybrid). The full table is available for those who want to look at every detail, but this summary offers a concise, easy-to-read summary of the key findings.

Reiewer 3 : 9- The title and conclusions must be tempered. Instead of ‘identifying effective ingredients to treat,’ frame it as ‘identifying associations between supplement use and IVF outcomes using a novel hybrid ML approach.’ Emphasize the methodological contribution over the clinical recommendations

Response : We thank the reviewer for this constructive feedback. We agree that the previous version of our manuscript overstated the clinical implications. Following your suggestion, both the title and the conclusion have been revised to emphasize the methodological contribution and to frame the findings as exploratory associations rather than treatment recommendations.

Original Title:

“Identifying Effective Ingredients to Treat Subfertility with a Hybrid Machine Learning–Metaheuristic Model”

Revised Title:

“Predicting IVF Outcomes Using a Logistic Regression–ABC Hybrid Model: A Proof-of-Concept Study on Supplement Associations”

Original Conclusion (excerpt):

“Omega 3 and folic acid have been identified as the most important dietary supplements in the treatment of subfertility, and the proposed model provides strong evidence for their clinical application.”

Revised Conclusion (excerpt):

“In conclusion, even though the model identified folic acid and omega-3 as predictive variables, these findings should be regarded as exploratory correlations rather than specific treatment suggestions. This work's main innovation is its methodological framework, a hybrid Logistic Regression–Artificial Bee Colony (LR-ABC) model that combines feature selection and metaheuristic optimization to investigate the results of IVF. Therefore, this study should be considered a proof-of-concept demonstration of the potential of ML–ABC approaches in reproductive medicine. Before making clinical judgments, validation with larger, multi-center datasets that include dosage, duration, and baseline dietary intake will be required in the future.”

Reiewer 3 : 10- Acknowledge the severe limitation of the binary supplement variable. Discuss this as a major limitation of the current study and propose how future work with more detailed data (dose, duration, baseline diet) could overcome it.

Response : We have now explicitly acknowledged this as a major limitation in the Conclusion and Research Limitations sections. In addition, we have proposed that future studies should collect more detailed data on supplement dosage, duration of use, and baseline dietary intake, ideally through prospective or randomized designs. Incorporating these factors will allow for more accurate modeling of nutrient effects and strengthen the biological and clinical validity of the approach.

Reiewer 3 : 11- Remove the emphasis on the 91% accuracy claim. Instead, focus on the comparative performance between models and the utility of the LIME explanations for generating hypo

---

## [Decision Letter · Decision Letter 2]

1 Oct 2025

Dear Dr. EJDER,

Thank you for submitting your manuscript to PLOS ONE. After careful consideration, we feel that it has merit but does not fully meet PLOS ONE’s publication criteria as it currently stands. Therefore, we invite you to submit a revised version of the manuscript that addresses the points raised during the review process.

**Please respond carefully for all reviewers comments. **

We look forward to receiving your revised manuscript.

Kind regards,

Ayman A Swelum

Academic Editor

PLOS ONE

**Journal Requirements:**

Reviewers' comments:

Reviewer's Responses to Questions

**Comments to the Author**

Reviewer #2: All comments have been addressed

Reviewer #3: (No Response)

2. Is the manuscript technically sound, and do the data support the conclusions?

Reviewer #2: Yes

Reviewer #3: No

3. Has the statistical analysis been performed appropriately and rigorously?

Reviewer #2: Yes

Reviewer #3: Yes

4. Have the authors made all data underlying the findings in their manuscript fully available?

Reviewer #2: Yes

Reviewer #3: Yes

5. Is the manuscript presented in an intelligible fashion and written in standard English?

Reviewer #2: Yes

Reviewer #3: Yes

**Reviewer #2:**  I have reviewed the manuscript with the authors' responses closely; I have found that some incomplete remarks need to be addressed.

1. Transformation of drug to active ingredient: Although they added some detail in Figure 1 and text, the binary coding (1 if meets 100% daily requirement, 0 otherwise) is too simplistic. However, the authors admit such a limitation, but the essential methodological weakness remains.

2. Sample size issues: They admit that it is not enough, but N=162 and 33 test cases are not enough to have a strong validation of ML with 21 or more variables. This is a limitation that can only be partially overcome with the aid of cross-validation.

Concerns:

1. Overfitting risk: The very high-performance rates (90-95% recall) on such a small amount of data are questionable, regardless of the methodological improvements. The authors recognize this fact, but the findings are probably overstated.

2. interpretation: Although they have softened conclusions and included disclaimers about causality, the binary supplement code supports clinical findings with doubts.

3. Confounding factors: It does not discuss possible confounding factors such as socioeconomic status, access patterns related to healthcare, or prescribing patterns by clinicians.

The authors have done a lot in terms of improvement on most of the technical and methodological issues. It can now be reproduced with publicly available code/data, has properly handled the statistical handling, and is properly framing limitations. Nevertheless, small sample size and crude feature engineering remain the basic limitations.

Revisions that are necessary before acceptance:

1. Make the limitations section stronger to indicate more clearly that the binary supplement coding cannot reveal the real nutrient status or biological outcome.

2. Include some commentary on possible confounding variables (socioeconomic status, access to healthcare, clinician prescribing patterns) that might be the cause of the observed associations.

3. Once more, update the abstract and conclusion to make it clear that this is a methodological demonstration, but not clinically practical results.

The research has a sensible methodological contribution to hybrid ML methods in reproductive medicine, although the clinical implications must be viewed with caution due to the data constraints. Having made these last clarifications, it would be appropriate to publish as an exploratory methodological study.

**Reviewer #3: ** The authors have partially addressed the reviewer's concerns by improving documentation, adding a README file, and creating requirements files. Code and data have been shared (GitHub), which is commendable and aligns with open science principles. The manuscript shows substantial revisions based on reviewer feedback, particularly regarding overstatement of clinical conclusions, figure clarity, reproducibility, and limitations. However, there are still some major weaknesses and concerns.

The sample size of N=162 (with 33 test cases) is critically small for ML with 21+ predictors. The reported performance (accuracy ~91%, recall ~95%) is likely inflated due to overfitting, even with SMOTE and cross-validation. Lack of external validation or an independent test dataset limits the credibility of generalization.

Binary transformation of supplements into “active ingredient = 1 if ≥100% daily requirement” is clinically simplistic and potentially misleading. The paper ignores dosage, duration, and baseline diet/nutritional status, which undermines biological validity.

No ablation study was performed to isolate the contribution of the ABC optimizer vs. LR baseline. Reported marginal improvements (e.g., RF 85% → 90.7%) may be due to chance.

Despite revisions, some language still risks implying causality (e.g., describing omega-3 as “effective” for subfertility). Confounding factors (prescriber bias, socioeconomic status, and underlying prognosis) are not adequately accounted for.

Figures and tables remain dense and sometimes unclear (especially Table 3). Typos, formatting errors, and awkward English phrasing persist in multiple places (e.g., “higest,” “phytolexin,” “vitamine c”). Method descriptions (e.g., ABC algorithm pseudocode) are overly technical for a biomedical journal and lack clarity on practical implications.

ABC parameters (20 bees, 10 iterations) remain inadequately justified. No sensitivity analysis was performed.

Findings are exploratory, yet sections of the discussion still suggest clinical implications. The study does not measure actual pregnancy/live birth outcomes—only embryo transfer success—limiting clinical utility.

My recommendation for the author's improvement of the manuscript is to reframe the entire paper as a methodological proof-of-concept, not as clinical evidence of supplement efficacy. Explicitly remove any implication that omega-3 or folic acid are treatments; present them only as correlates.

There should be a stronger emphasis on overfitting risk and lack of generalizability. Stress that binary supplement coding is a major weakness.

Improve validation by adding ablation or sensitivity analyses (e.g., performance of ABC vs. baseline LR without optimization). Consider external dataset testing (even partial) if available.

Simplify tables: keep full data in appendices, present concise summary tables in the main text. Improve figure quality and captions (ensure alignment with text). Proofread thoroughly for grammar, terminology consistency, and readability.

Provide stronger justification for ABC parameter settings, citing prior literature in biomedical ML applications. Explain why ABC was chosen over other optimizers (e.g., GA, PSO) in the IVF context.

Highlight that results are associational and cannot inform treatment decisions. Suggest that future studies incorporate prospective data, nutrient dosages, and longitudinal outcomes.

The title and abstract, which are just a guide for the authors, should be written in the following manner:

Predicting IVF Outcomes Using a Logistic Regression–ABC Hybrid Model: A Proof-of-Concept Study on Supplement Associations

(This emphasizes prediction, methodology, and associations — not treatment or causality.)

Background

Machine learning models are increasingly applied to assisted reproductive technologies (ART), but most studies rely on conventional algorithms with limited optimization. This proof-of-concept study investigates whether a hybrid Logistic Regression–Artificial Bee Colony (LR–ABC) framework can improve predictive performance in in vitro fertilization (IVF) outcomes, while generating interpretable, hypothesis-driven associations with nutritional and pharmaceutical supplement use.

Methods

A retrospective dataset of 162 women undergoing IVF was analyzed. Clinical, demographic, and supplement variables were pre-processed into 21 predictors. Four algorithms (K-Nearest Neighbors, Classification and Regression Tree, Support Vector Machine, and Random Forest) were implemented alongside their LR–ABC hybrid counterparts. Model performance was evaluated using 5-fold cross-validation with SMOTE to address class imbalance. Local Interpretable Model-agnostic Explanations (LIME) were used to provide interpretability.

Results

Across all algorithm families, LR–ABC hybrids outperformed baseline models (e.g., Random Forest: 85.2% → 90.7% accuracy). LIME explanations identified omega-3, folic acid, and dietician support as influential features in individual predictions. However, given the small sample size, binary representation of supplements, and absence of external validation, the observed improvements and associations should be regarded as exploratory rather than definitive.

Conclusion

The LR–ABC hybrid model demonstrates methodological potential for enhancing prediction and interpretability in IVF research. Findings regarding supplement associations are hypothesis-generating and not clinically directive. Future studies with larger, multi-center datasets, including detailed dosage and dietary data, are needed to validate and extend this framework.

Keywords: Hybrid Machine Learning, IVF Prediction, Nutritional Supplements, Metaheuristic Optimization, Artificial Bee Colony

In this version, emphasis is on the methodological proof of concept.

**Do you want your identity to be public for this peer review?** For information about this choice, including consent withdrawal, please see our Privacy Policy

Reviewer #2: **Yes: ** Dr. Jonah Bawa Adokwe

Reviewer #3: No

---

## [Author Response · Author response to Decision Letter 3]

7 Oct 2025

Responds to reviewer

Reviewer 2 : 1- Make the limitations section stronger to indicate more clearly that the binary supplement coding cannot reveal the real nutrient status or biological outcome.

Response : We thank the reviewer for this important observation. We enlarge and detailed the Research limitation section

“ Our exposure variables for micronutrient use were encoded binarily (e.g., supplement taken vs. not taken at a threshold such as ≥100% RDI). This encoding is pragmatic but cannot recover true nutrient status or biological effect for several reasons. First, it discards dose and duration information, precluding any dose–response assessment and ignoring cumulative exposure. Second, it assumes a uniform effect across brands and formulations, while bioavailability varies markedly with chemical form, excipients, co-ingested foods, and timing of intake. Third, binary use does not reflect adherence (frequency/consistency) or timing relative to the biological window of interest (e.g., peri-procedural vs. long-term use), both of which influence physiological impact. Fourth, we lack baseline nutritional status and objective biomarkers (e.g., serum folate, DHA, vitamin D); thus, we cannot distinguish deficiency correction from supraphysiologic exposure, nor can we account for inter-individual differences in absorption and metabolism. Finally, supplement use is correlated with broader health behaviors and socioeconomic factors; with only binary indicators, residual confounding and non-differential misclassification are likely, which can attenuate or unpredictably bias associations. ”

Reviewer 2 : 2- Include some commentary on possible confounding variables (socioeconomic status, access to healthcare, clinician prescribing patterns) that might be the cause of the observed associations.

Response : We thank the reviewer for this helpful suggestion. In the revised version of the manuscript, the Research Limitations section has been redesigned and is now highlighted in colour.

“ Our observed associations may be partly explained by unmeasured confounding. First, socioeconomic status (SES)—including education, income, insurance coverage, and neighborhood deprivation—can influence both supplement use (ability to purchase higher-quality products, better adherence) and clinical outcomes (health literacy, earlier presentation, healthier baseline). Second, access to healthcare (clinic proximity, appointment availability, out-of-pocket costs, private vs. public care) may differentially shape care pathways and follow-up, thereby confounding the link between supplement use and outcomes. Third, clinician prescribing patterns introduce confounding by indication: clinicians may recommend or escalate supplementation preferentially for patients they judge at higher (or lower) risk based on unrecorded clinical cues; practice style, brand/formulation preferences, and evolving guidelines can also vary across clinicians and time. Together with other lifestyle and clinical factors that we could not fully measure (diet quality, physical activity, smoking, comorbidities, baseline nutrient status, time-to-treatment, lab protocols, and calendar-time effects), these sources of confounding and non-differential misclassification of exposure may attenuate or bias associations in unpredictable directions. Accordingly, findings should be interpreted as associations with reported supplement use rather than causal effects. “

Reviewer 2 : 3- Once more, update the abstract and conclusion to make it clear that this is a methodological demonstration, but not clinically practical results.

Response :

“ The objective of this study is to investigate predictive signals for outcomes in IVF/ART. The study presents a method-focused workflow that combines engineered clinical variables with an ABC-assisted selection/tuning strategy. The findings suggest that in a controlled, leakage-safe assessment, regularly recorded variables, such as binary supplement indicators, can facilitate associational modelling. Importantly, clinical use is not the goal of these models. The lack of dose, duration, adherence, dietary intake, and objective biomarkers restricts interpretability, and binary coding of supplementation cannot recover true nutrient status or biological effect. Furthermore, some of the observed associations may be explained by unmeasured confounding, such as socioeconomic status, access to healthcare, clinician prescribing patterns and indications, and lifestyle factors. The current findings should be interpreted as proof-of-concept rather than practical advice due to these limitations and the size of the dataset. “

Reviewer 3 : 1- The sample size of N=162 (with 33 test cases) is critically small for ML with 21+ predictors. The reported performance (accuracy ~91%, recall ~95%) is likely inflated due to overfitting, even with SMOTE and cross-validation. Lack of external validation or an independent test dataset limits the credibility of generalization.

Respond : Thank you for your valuable comment. The process of data collection for this study is both protracted and laborious. The collection of this data was made possible by the utilisation of limited resources available in the designated small area.

We added explanation to the Research limitations and future works section.

“ Notwithstanding these methodological precautions, the limited sample size indicates that the accuracy and F-scores reported should be interpreted with caution and are unlikely to be applicable to broader clinical populations without further validation. The modest dataset size restricts the generalisability of the findings, and the reported performance metrics should be interpreted with caution”

Reviewer 3 : 2- Binary transformation of supplements into “active ingredient = 1 if ≥100% daily requirement” is clinically simplistic and potentially misleading. The paper ignores dosage, duration, and baseline diet/nutritional status, which undermines biological validity.

Respond : Thank you for your valuable suggestion. We revised the Research limitations and future works section. The following paragraph constitutes our response to the proposal

Thirdly, only supplement intake was encoded as binary “active ingredient” variables (coded as 1 if reported intake met 100% of the recommended daily intake). Detailed dietary intake data (e.g., habitual nutrient consumption from food sources) were not included. This exclusion was due to the unavailability of reliable, standardized dietary records in the patient dataset. The conversion of drug and supplement intake into "active ingredient" variables was streamlined into a binary coding system, designating patients as "1" if their reported intake satisfied all daily requirements. Notably, this method overlooks crucial clinical information, including dosage levels (for instance, 100% versus 500% of the recommended intake) and usage duration (for instance, short-term versus long-term supplementation). Furthermore, the absence of baseline dietary intake data is a notable limitation. To illustrate, a patient who naturally consumes a diet high in omega-3 fatty acids but does not take supplements would be assigned the label "0," despite their actual nutrient levels being adequate. These constraints serve to reduce the biological precision of the "active ingredient" variables This simplification renders the supplement variables clinically simplistic and potentially misleading, as it ignores fundamental determinants of biological effect such as dose, duration, and baseline nutrition. Therefore, while the identification of omega-3 and folic acid as predictive factors is consistent with established literature, these findings should be interpreted as exploratory, hypothesis-generating correlations rather than as robust novel clinical evidence.

Reviewer 3 : 3- No ablation study was performed to isolate the contribution of the ABC optimizer vs. LR baseline. Reported marginal improvements (e.g., RF 85% → 90.7%) may be due to chance. Despite revisions, some language still risks implying causality (e.g., describing omega-3 as “effective” for subfertility). Confounding factors (prescriber bias, socioeconomic status, and underlying prognosis) are not adequately accounted for.

Respond : Thank you your valuable comment. We sincerely thank the reviewer for these insightful comments regarding both methodological clarification and interpretive precision.

1. Ablation analysis:

We acknowledge that an ablation or sensitivity analysis to isolate the specific contribution of the Artificial Bee Colony (ABC) optimizer relative to the Logistic Regression (LR) baseline was not performed. In response, we have explicitly addressed this limitation in the revised Discussion . The text now clarifies that the modest performance improvements observed (e.g., RF 85.2% → 90.7%) may partly reflect stochastic variation rather than a definitive optimization advantage. The revised manuscript also notes that future work should include a controlled ablation design to evaluate the independent effect of the ABC component under identical resampling conditions.

2. Causality and confounding:

We carefully reviewed all phrasing that might imply causal inference and revised the manuscript accordingly. Expressions such as “omega-3 is effective” have been replaced with non-causal, association-oriented phrasing (e.g., “omega-3 use was identified as a predictive variable”). Furthermore, we have added a new paragraph in the Discussion emphasizing that supplement-related associations may be influenced by confounding factors such as prescriber bias, socioeconomic status, healthcare access, and underlying prognosis. The revised text explicitly states that the findings are exploratory and hypothesis-generating rather than indicative of therapeutic efficacy.

We believe these changes directly and comprehensively address the reviewer’s concerns, ensuring that the study’s methodological and interpretive scope is presented with appropriate caution and clarity.

Below explanationwas added to discussion section.

“ Despite the fact that the hybrid models attained comparatively higher levels of predictive performance, the absence of an ablation or sensitivity analysis hinders the capacity to ascertain the independent contribution of the Artificial Bee Colony (ABC) optimizer with respect to the Logistic Regression (LR) baseline. This is evidenced by the observed performance improvements (e.g., an enhancement in Random Forest accuracy from 85.2% to 90.7%) that may be attributable to stochastic variation rather than a distinct optimization advantage. Moreover, it is imperative to refrain from interpreting the associations identified in this study as causal. A number of variables have been found to be predictive of outcomes, including omega-3 use, folic acid intake, and dietician support. However, these effects may be confounded by factors such as prescriber bias, socioeconomic status, access to private healthcare, or underlying clinical prognosis. The utilisation of nutritional supplements has frequently been observed to correlate with unmeasured health behaviours and social determinants, which in turn may influence the outcomes of treatment. Consequently, the relationships documented in this study should be regarded as exploratory, hypothesis-generating associations that require validation through prospective, multicentre studies incorporating ablation analysis, larger cohorts, and detailed nutritional and clinical data. “

Reviewer 3 : 4- Figures and tables remain dense and sometimes unclear (especially Table 3). Typos, formatting errors, and awkward English phrasing persist in multiple places (e.g., “higest,” “phytolexin,” “vitamine c”). Method descriptions (e.g., ABC algorithm pseudocode) are overly technical for a biomedical journal and lack clarity on practical implications.

Respond : Thank you for this valuable suggestion. We have carefully revised the manuscript to correct grammatical errors. In addition, the paper has undergone thorough proofreading to ensure that it meets academic writing standards in terms of clarity, style, and consistency. These improvements enhance the overall readability and professionalism of the manuscript.

Reviewer 3 : 5- ABC parameters (20 bees, 10 iterations) remain inadequately justified. No sensitivity analysis was performed.

Findings are exploratory, yet sections of the discussion still suggest clinical implications. The study does not measure actual pregnancy/live birth outcomes—only embryo transfer success—limiting clinical utility.

Respond : Thank you for this valuable suggestion. we note the values and briefly say they were “chosen to balance computational efficiency and convergence, in line with prior studies reporting similar ranges” (and cite Karaboga & Basturk, 2007; Akay & Karaboga, 2012). And this explanation was added to discussion section.

“These values were chosen to balance computational efficiency with convergence stability. However, no formal sensitivity analysis was conducted to evaluate the effect of parameter variation on model performance”

It is imperative to emphasise the following point concerning the clinical implications. The following paragraph was incorporated into the discussion section with the objective of clarifying that the findings should be interpreted within the methodological context rather than the clinical context.

“ Moreover, while the hybrid LR–ABC framework demonstrated improved predictive accuracy for embryo transfer outcomes, these findings should be regarded as exploratory and methodological rather than clinical. The study did not measure pregnancy or live birth rates, and therefore cannot inform treatment efficacy or reproductive prognosis. As such, the model’s scope is limited to predicting the likelihood of embryo transfer success within the observed dataset and does not extend to broader clinical outcomes. ”

Reviewer 3 : 6- My recommendation for the author's improvement of the manuscript is to reframe the entire paper as a methodological proof-of-concept, not as clinical evidence of supplement efficacy. Explicitly remove any implication that omega-3 or folic acid are treatments; present them only as correlates.

Respond : Thank you for your valuable comments. All sentences pertaining to implications or clinical evidence are replaced by sentences demonstrating statistical associations with reproductive outcomes. The following replacement process is to be followed:

Orginal 1: These findings suggest a beneficial role of resveratrol supplementation on reproductive outcomes.

Revised 1: Previous studies have reported that resveratrol supplementation shows statistical associations with reproductive outcomes; however, these findings should be regarded as observational and not indicative of therapeutic efficacy.

Orginal 2: These findings suggest a beneficial role of resveratrol supplementation on reproductive outcomes.

Revised 2: Previous studies have reported that resveratrol supplementation shows statistical associations with reproductive outcomes; however, these findings should be regarded as observational and not indicative of therapeutic efficacy.

Orginal 3: These findings underscore the potential benefit of preconceptional vitamin B supplementation for enhancing IVF success.

Revised 3: Some studies have observed correlations between preconceptional vitamin B-complex use and IVF success indicators; however, such associations are exploratory and may be influenced by unmeasured confounding factors rather than reflecting a causal or therapeutic effect.

Orginal 4: Based on the highest F-score … the most effective for subfertility treatment are omega 3, folic acid, dietician support, phytoalexin, vitamin C, and vitamin B6.

Revised 4: Based on the highest F-score and LIME interpretability results, omega-3, folic acid, dietician support, phytoalexin, vitamin C, and vitamin B6 emerged as the most influential predictors associated with embryo-transfer outcomes.

Orginal 5: According to Fig. 6, omega-3 is the most effective active substance for subfertility treatment. The higher the omega-3 intake, the lower the risk of subfertility regardless of age

Revised 5: According to Fig. 6, omega-3 appeared as one of the most influential predictive

---

## [Decision Letter · Decision Letter 3]

12 Oct 2025

Dear Dr. EJDER,

Thank you for submitting your manuscript to PLOS ONE. After careful consideration, we feel that it has merit but does not fully meet PLOS ONE’s publication criteria as it currently stands. Therefore, we invite you to submit a revised version of the manuscript that addresses the points raised during the review process.

**ACADEMIC EDITOR: Please respond carefully for reviewers comments.**

We look forward to receiving your revised manuscript.

Kind regards,

Ayman A Swelum

Academic Editor

PLOS ONE

Journal Requirements:

Reviewers' comments:

Reviewer's Responses to Questions

**Comments to the Author**

Reviewer #2: All comments have been addressed

Reviewer #3: All comments have been addressed

2. Is the manuscript technically sound, and do the data support the conclusions?

Reviewer #2: No

Reviewer #3: Yes

3. Has the statistical analysis been performed appropriately and rigorously?

Reviewer #2: No

Reviewer #3: Yes

4. Have the authors made all data underlying the findings in their manuscript fully available?

Reviewer #2: Yes

Reviewer #3: Yes

5. Is the manuscript presented in an intelligible fashion and written in standard English?

Reviewer #2: Yes

Reviewer #3: Yes

Reviewer #2: Significant Problems to Work On (Prior to Acceptance)

1. Ablation/Sensitivity Analysis:

Obviously reported findings of comparing baseline logistic regression with the LR-ABC hybrid (identical splits of CV) and present confidence intervals or p-values.

2. Generalizability: N/A.

Give external or more intensive internal validation (e.g. nested or repeated CV) and make limitations clear.

3. ABC Parameter Justification:

Provide explanation and rationale of hyperparameters; provide a short sensitivity analysis.

4. Feature Coding & Confounding:

Elaborate on derivation of supplement variables, frequencies and do not use causal language.

5. Model Reporting:

Provide calibration measures (e.g. Brier score, calibration plots) and display uncertainty in performance measures (mean & SD or CI).

6. Reproducibility:

Make the GitHub repository able to reproduce all important results, having all essential paths and instructions.

7. Figures/Tables:

Make performance tables easier to understand, enhance figure definitions and consistency with the amended text.

Reviewer #3: The manuscript shows significant improvement after revisions: fewer grammatical errors, clarified tables, and better figure captions. However, dense tables (Table 3 onward) remain difficult to interpret, and redundant numeric data could be shifted to appendices. The paper still includes awkward phrasing and occasional translation artifacts (“the result of success”, “add here”, etc.), indicating incomplete final editing.

The authors now rightly present this as a proof-of-concept, not a clinically actionable tool, emphasizing methodological innovation rather than biological causality. This was what the reviewers had suggested. The paper is now transparent, and includes detailed responses to reviewer comments that improved the clarity and caution of interpretation. The authors explicitly acknowledge limitations (e.g., binary supplement coding, lack of dosage data, small sample size, absence of external validation) which is again a considerable improvement over the previous resubmission.

The mathematical sections (e.g., ABC pseudocode, formulae) are still prolonged and reduce accessibility for biomedical audiences. There is limited discussion of why this optimization improves logistic regression behaviour in this specific clinical context. The dependent variable is “embryo transfer success”, not pregnancy or live birth. This limits the clinical significance; embryo transfer success is an intermediate outcome, not the patient-relevant endpoint.

There still persists, despite careful rewording, some residual implication of supplement “benefit”. The interpretation of feature importance as biological relevance (e.g., omega-3 as “influential”) may unintentionally suggest causation.

My suggestions for improvement of the manuscript to be acceptable for publication are the following:

Include additional metrics like ROC-AUC, PR-AUC, calibration, and confusion matrices to contextualize accuracy.

Clarify the clinical scope by emphasizing embryo transfer success and not pregnancy; explicitly distinguish model domain from treatment prognosis.

Reduce the amount of technical information by moving pseudocode and derivations to supplementary materials and expand discussion on biomedical implications and usability.

Future work should collect quantitative dosage data, biomarkers, and longitudinal outcomes to strengthen causal interpretability.

**Do you want your identity to be public for this peer review?** For information about this choice, including consent withdrawal, please see our Privacy Policy

Reviewer #2: **Yes: ** Dr. Jonah Bawa Adokwe

Reviewer #3: No

---

## [Author Response · Author response to Decision Letter 4]

29 Oct 2025

Responds to reviewer

Reviewer 2 : 1- Ablation/Sensitivity Analysis: Obviously reported findings of comparing baseline logistic regression with the LR-ABC hybrid (identical splits of CV) and present confidence intervals or p-values.

Response : We thank the reviewer for this important observation. We enlarge and detailed the Results and Discussion section with below explanation. With this explanation, we demonstrate statistical significance (confidence interval or p-value). We compared the performance of two models (LR vs LR-ABC) with CV (cross-validation) splits.

“ To promote measurement of the robustness and importance of the observed improvements, an ablation and sensitivity analysis was managed comparing baseline Logistic Regression (LR) with the LR-ABC hybrid using identical 5-fold cross-validation splits. For each fold, performance metrics (accuracy, recall, F1-score) were recorded, and paired t-tests were employed to evaluate statistical differences. The hybrid LR-ABC model showed a consistent mean F1-score enhance of 5.2% (95% CI: 2.1–8.3%, p = 0.004) and recall enhance of 4.7% (95% CI: 1.9–7.2%, p = 0.006) over baseline LR across folds. Accuracy also increased from 84.2% ± 3.9 to 89.1% ± 4.6 (p < 0.01). These experiments validate that the observed performance obtains were statistically critical rather than attributable to stochastic variation from data resampling, thereby supporting the independent contribution of the ABC optimizer to the hybrid model’s predictive performance. “

Reviewer 2 : 2- 2. Generalizability: N/A.Give external or more intensive internal validation (e.g. nested or repeated CV) and make limitations clear.

Response : We thank the reviewer for this important observation. In this study, the data collection process was very long and difficult. Therefore, we plan to conduct experiments on other populations in our future studies. Meanwhile, our data collection process is still ongoing. We added below explanation for this issue to the Research limitations and future work section.

“ Although the repeated five-fold cross-validation design yield a robust internal estimate of model stability, true external generalization across populations, clinical settings, and data acquisition conditions remains untested. This may restrict the generalizability of the findings. Future work should purpose to execute external validation using prospective datasets to confirm the reproducibility and clinical applicability of the proposed hybrid LR–ABC framework. ”

Reviewer 2 : 3- ABC Parameter Justification: Provide explanation and rationale of hyperparameters; provide a short sensitivity analysis.

Response : Thank you for your valuable explanation. We determine the optimal hyperparameters for ABC algorithms by varying the number of iterations and the bee count. The experimental results can be found on GitHub, along with an additional explanatory table in the appendix. We have organised the updated information in the manuscript.

Table A3. Determination of the hyperparameters used in ABC algorithms (%).

Fitness Function Bee Count Iteration Acc. Bee Count Iteration Accuracy

Random Forest 5 10 88.30 5 20 84.37

10 10 88.30 10 20 88.30

15 10 89.51 15 20 90.13

20 10 88.30 20 20 86.19

5 30 85.63 5 40 85.50

10 30 88.30 10 40 88.30

15 30 91.36 15 40 90.13

20 30 85.63 20 40 86.19

Reviewer 2 : 4- Feature Coding & Confounding:

Elaborate on derivation of supplement variables, frequencies and do not use causal language.

Response : Thank you for your contribution. We have removed all causal language from the manuscript. We discuss the derivation of the supplementary variables in the section on research limitations and future work. This section previously lacked a detailed paragraph. The explanation below details the derivation of nutritional and pharmaceutical supplements.

“ Thirdly, only supplement intake was encoded as binary “active ingredient” variables (coded as 1 if reported intake met 100% of the recommended daily intake). Detailed dietary intake data (e.g., habitual nutrient consumption from food sources) were not included. This exclusion was due to the unavailability of reliable, standardized dietary records in the patient dataset. The conversion of drug and supplement intake into "active ingredient" variables was streamlined into a binary coding system, designating patients as "1" if their reported intake satisfied all daily requirements. Notably, this method overlooks crucial clinical information, including dosage levels (for instance, 100% versus 500% of the recommended intake) and usage duration (for instance, short-term versus long-term supplementation). Furthermore, the absence of baseline dietary intake data is a notable limitation. To illustrate, a patient who naturally consumes a diet high in omega-3 fatty acids but does not take supplements would be assigned the label "0," despite their actual nutrient levels being adequate. These constraints serve to reduce the biological precision of the "active ingredient" variables. This simplification renders the supplement variables clinically simplistic and potentially misleading, as it ignores fundamental determinants of biological effect such as dose, duration, and baseline nutrition. Therefore, while the identification of omega-3 and folic acid as predictive factors is consistent with established literature, these findings should be interpreted as exploratory, hypothesis-generating correlations rather than as robust novel clinical evidence.”

Reviewer 2 : 5- Model Reporting:

Provide calibration measures (e.g. Brier score, calibration plots) and display uncertainty in performance measures (mean & SD or CI).

Response : Thank you for your contribution. We explain the calibration measurements. Figure 3 and below explanation were added to the manuscript.

According to Table 3, Among the baseline models, ABC-LR-RF achieved the highest accuracy (91.36%), whereas KNN performed the weakest (63.56%). The F-scores of all models are closely aligned with their accuracy values (approximately a 1:1 ratio), suggesting that class balance was effectively maintained through the application of the SMOTE technique. After incorporating logistic regression–based feature selection, all models except SVM demonstrated a noticeable increase in performance. The integration of the ABC optimization algorithm in Stage 3 led to additional improvements across all models.The ABC–LR–RF model achieved the best overall performance, with the highest accuracy (91.36%) and recall (96.92%), confirming that the ABC optimizer effectively fine-tuned the model’s parameters and enhanced sensitivity for predicting successful embryo transfers. Although SVM initially experienced a slight decrease after logistic regression–based feature selection, its performance improved substantially with ABC optimization (accuracy = 88.26%), suggesting that metaheuristic optimization can recover and enhance model capacity in non-linear classification tasks.

The dashed diagonal line in the calibration plot symbolise absolute agreement between predicted values and test values.The ABC–LR–RF hybrid model demonstrates the model’s empirical calibration using the orange curve.Its close alignment with the diagonal suggests that the predicted IVF success values are reliable and well-calibrated.

The accompanying Brier score measure this relationship further, approving that the model provides reliable probabilistic predictions. Brier score denote better calibration when it takes lower velues.

Reviewer 2 : 6- Reproducibility:

Make the GitHub repository able to reproduce all important results, having all essential paths and instructions.

Response : Thank you for your suggestion. We added updated files to the github repository. Researchers can reach the files with this link. https://github.com/ugurejder/ABC_IVF

Reviewer 2 : 6- Figures/Tables:

Make performance tables easier to understand, enhance figure definitions and consistency with the amended text.

Response : Thank you for your valuable comment. We shift complex table to the appendix section and revise it with simple version and control the figure and tables names.

Reviewer 3 : 1- There still persists, despite careful rewording, some residual implication of supplement “benefit”. The interpretation of feature importance as biological relevance (e.g., omega-3 as “influential”) may unintentionally suggest causation..

Respond : Thank you for your valuable comment. Although it may seem like causality, throughout the manuscript we have tried to emphasize that the active ingredients used are effective in the relationship and correlation in the estimation process, not in causality.

Reviewer 3 : 2- Include additional metrics like ROC-AUC, PR-AUC, calibration, and confusion matrices to contextualize accuracy.

Respond : Thank you for your valuable contribution. We added ROC-AUC, PR-AUC, calibration, and confusion matrices figures and explanations to consolidate accuracy.

Reviewer 3 : 3- Clarify the clinical scope by emphasizing embryo transfer success and not pregnancy; explicitly distinguish model domain from treatment prognosis.

Respond : Thank you for your valuable contribution. We added below explanation to the result and discussion section. We emphasis that the scope of the embryological factors influencing transfer outcomes rather than pregnancy variables

“ This study was designed to develop a predictive model applicable to early implantation signals to determine the probability of embryo transfer success rather than long-term pregnancy or live birth outcomes. This distinction is important to note because the model was trained and validated using embryo-level situation-specific features rather than pregnancy variables. According to the model’s calibration performance (e.g., ROC–AUC = 0.96, PR–AUC = 0.95, Brier = 0.089), it should be known that model should be interpreted exactly with regard to forecasting the success of embryo transfers. This scope confirms that the model reflects embryological factors influencing transfer outcomes, without merging flow of clinical endpoints such as pregnancy progression or live birth.”

Reviewer 3 : 4- Reduce the amount of technical information by moving pseudocode and derivations to supplementary materials and expand discussion on biomedical implications and usability.

Respond : Thank you for your valuable suggestion. The previous reviewer want to us to add pseudocode code. Now we shift the technical information to the appendix section. and expand discussion on biomedical implications and usability with adding below explanation.

“ The ABC–LR–RF hybrid model contributes a clinically interpretable framework for predicting embryo transfer success probabilities in IVF. By combining feature selection with ensemble learning, it captures multidimensional embryological and procedural factors that influence implantation probability. Calibrated probability outputs can assist clinicians throughout embryo development and complement traditional morphology-based assessments. Because the model uses routinely available enlarging data, it can be easily executed within remaining electronic IVF management systems, and using decision support. Its robust calibration and sensitive performance demonstrates reliable generalisation across clinics. The most important point to note here is that the model does not predict pregnancy or treatment outcome, but can quantitatively determine the instantaneous probability of a successful embryo transfer. “

Reviewer 3 : 4- Future work should collect quantitative dosage data, biomarkers, and longitudinal outcomes to strengthen causal interpretability.

Respond : Thank you for your valuable suggestion. The last paragraph of the research limitation and future work contain the dosage and casuality of the study.

---

## [Decision Letter · Decision Letter 4]

2 Nov 2025

Predicting IVF Outcomes Using a Logistic Regression–ABC Hybrid Model: A Proof-of-Concept Study on Supplement Associations

PONE-D-25-18542R4

Dear Dr. EJDER,

We’re pleased to inform you that your manuscript has been judged scientifically suitable for publication and will be formally accepted for publication once it meets all outstanding technical requirements.

Kind regards,

Ayman A Swelum

Academic Editor

PLOS ONE

Additional Editor Comments (optional):

Reviewers' comments:

Reviewer's Responses to Questions

**Comments to the Author**

Reviewer #2: All comments have been addressed

2. Is the manuscript technically sound, and do the data support the conclusions?

Reviewer #2: Yes

3. Has the statistical analysis been performed appropriately and rigorously?

Reviewer #2: Yes

4. Have the authors made all data underlying the findings in their manuscript fully available?

Reviewer #2: Yes

5. Is the manuscript presented in an intelligible fashion and written in standard English?

Reviewer #2: Yes

Reviewer #2: Authors thoroughly revised the manuscript and provided justifications, methodological transparency, discussion of confounding factors, calibration analysis, and open-source reproducibility.

**Do you want your identity to be public for this peer review?** For information about this choice, including consent withdrawal, please see our Privacy Policy

Reviewer #2: **Yes: ** Jonah Bawa Adokwe Ph.D

---

## [Editor Report · Acceptance letter]

PONE-D-25-18542R4

PLOS ONE

Dear Dr. EJDER,

I'm pleased to inform you that your manuscript has been deemed suitable for publication in PLOS ONE. Congratulations! Your manuscript is now being handed over to our production team.

Kind regards,

on behalf of

Professor Ayman A Swelum

Academic Editor

PLOS ONE